statistics/environmental science

Extreme value analysis, generalized Pareto distribution, maximum yield levels, winter wheat yield

**Author for correspondence:**
Emily G. Mitchell
e-mail: emily.mitchell@nottingham.ac.uk

# Operating at the extreme: estimating the upper yield boundary of winter wheat production in commercial practice

Emily G. Mitchell[1], Neil M. J. Crout[2], Paul Wilson[2], Andrew T. A. Wood[3] and Gilles Stupfler[1,4]

[1]School of Mathematical Sciences, University of Nottingham, Nottingham NG7 2RD, UK
[2]School of Biosciences, University of Nottingham, Sutton Bonington LE12 5RD, UK
[3]Research School of Finance, Actuarial Studies and Statistics, Australian National University, 26C Kingsley Street, ACT 2601 Australia
[4]Univ Rennes, Ensai, CNRS, CREST - UMR 9194, 35000 Rennes, France

EGM, 0000-0002-2581-0099; NMJC, 0000-0001-7394-5070;
PW, 0000-0003-4802-4127; ATAW, 0000-0003-2975-1986;
GS, 0000-0003-2497-9412

Wheat farming provides 28.5% of global cereal production. After steady growth in average crop yield from 1950 to 1990, wheat yields have generally stagnated, which prompts the question of whether further improvements are possible. Statistical studies of agronomic parameters such as crop yield have so far exclusively focused on estimating parameters describing the whole of the data, rather than the highest yields specifically. These indicators include the mean or median yield of a crop, or finding the combinations of agronomic traits that are correlated with increasing average yields. In this paper, we take an alternative approach and consider high yields only. We carry out an extreme value analysis of winter wheat yield data collected in England and Wales between 2006 and 2015. This analysis suggests that, under current climate and growing conditions, there is indeed a finite upper bound for winter wheat yield, whose value we estimate to be 17.60 tonnes per hectare. We then refine the analysis for strata defined by either location or level of use of agricultural inputs. We find that there is no statistical evidence for variation of maximal yield depending on location, and neither is there statistical evidence that maximum yield levels are improved by high levels of crop protection and fertilizer use.

# 1. Introduction

Wheat is one of the most important food crops in the world. Current global annual production levels of wheat stand at 756.8 million tonnes [1], two-thirds of which is used for human consumption in food staples such as bread. As a result of sustained improvements to crop varieties and agricultural technology, there was a progressive and very large increase in wheat yields over the second half of the last century [2]. Despite this, there are concerns for the future growth of crop yield, the main one arguably being climate change. Recent literature has focused on forecasting the behaviour of crops in a changing climate [3–6], and found that a global temperature increase may lead to a yield reduction in cereal crops in certain regions. At the same time, current projections point to major increases in demand for food and livestock feed, as well as rising demand for biofuels due to a progressive shift of major economic powers to generating energy via renewable sources [7]. Understanding the drivers of crop yield, and specifically quantifying the upper bound of yield, is of crucial importance to successfully address the challenge of global food security.

We tackle this question with the example of agricultural production in the UK, where wheat is the most widely grown arable crop [8], and the most planted variety of wheat is winter wheat (or *Triticum aestivum*). UK wheat yields have risen from a little over 2 tonnes per hectare in the early twentieth century [9] to current averages of approximately 8 tonnes per hectare [8]. It is, however, apparent that average UK wheat yields have stagnated over the last 20 years [10], even though the understanding of climate mechanisms and biotechnology has made huge progress over this period. The phenomenon of stagnant average wheat yield is not limited to the UK; it has, for instance, been noted in continental Europe as well [11]. Based on this observation, one may wonder whether wheat yields have reached a maximal or near-maximal level; in any case, substantial variation in observed wheat yield levels exists, and in the context of food security we seek to estimate the maximum achievable yield of winter wheat under current technologies and growing conditions.

We address this question using extreme value analysis, which is a statistical framework used to model the atypically high events which only occur with a very small probability. Extreme value theory has found applications in numerous fields, the most prominent examples being environmental science [12–14] and insurance and finance [15–17]. Other applications include engineering [18,19] and toxicology [20]. More recently, extreme value analysis has been used in epidemiology to estimate the probability of severe pneumonia and influenza epidemics [21], and in the field of demography with a discussion of whether there is a finite upper bound on human lifespan [22]. Applications of extreme value analysis in the agricultural sciences have so far concentrated on financial aspects, for instance commodity price fluctuations [23,24], rather than agronomic factors such as yield. The focus of the applied statistical literature on understanding agricultural yield variability has typically been to estimate average yield levels and understand the relationships that drive them using central, rather than extreme, statistical methodology such as principal component and path coefficient analyses [25–28].

In our context, the extreme value analysis of yields stands for modelling the highest yields. We carry out this analysis using data on winter wheat yields collected in England and Wales by the Farm Business Survey (FBS) between 2006 and 2015. Let us highlight here that our objective is not to estimate the notion of yield potential, which is equal to the yield of a crop under ideal conditions (no pest, disease, nutrient or water stresses), or the related notion of water-limited yield potential [29,30]. The estimation of these quantities typically requires the use of sophisticated computer models to simulate crop growth in specified conditions [31,32]; our goal is rather to estimate the distribution of the highest yields attained in a real-world setting and under observed farming practices in order to estimate a practical upper bound on yield given current technology and conditions.

Our analysis of the highest wheat yields can also be refined to take growing conditions into account. In the literature, forecasts for winter wheat yields have been calculated for geographical regions, such as the Nomenclature of Territorial Units for Statistics level 1 (NUTS1) regions in Germany and France [33] or administrative regions in the UK defined by the Met Office [34]. This makes it possible to assess the variation in yield levels depending on climate and practices. The effect of the use of agricultural inputs, mainly fertilizers and crop protection, on average yield levels is also of interest; it is important, in this respect, to assess the trade-off between an improvement in yield and potential damage to the environment that may result from excessive use of those inputs. It has thus been found in the literature that the use of crop protection and fertilizer does indeed generally improve yield, but that a moderate level of these inputs typically brings the same improvement as higher levels without incurring the same risks to the environment and human health [35–37]. Using further information contained in the FBS database, we carry out extreme value analyses of winter wheat yield depending on location and level of crop protection and fertilizer use. We then compare the conclusions of each of these analyses, and contrast them with the interpretation of the extreme value analysis of the full, non-stratified sample of yields.

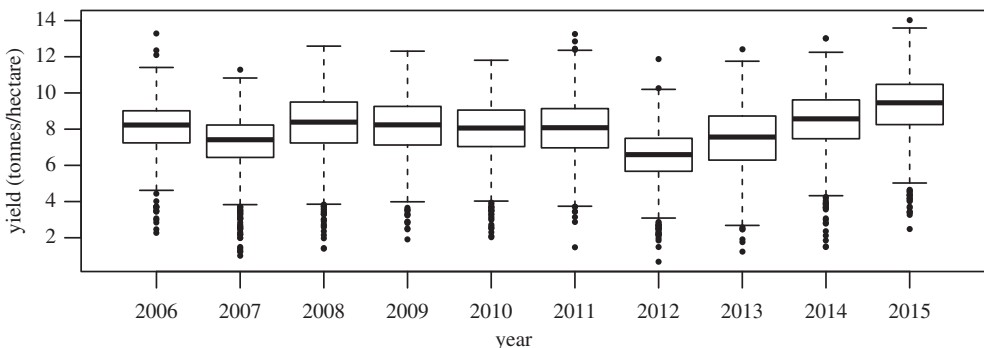

**Figure 1.** Annual yield boxplots using the FBS data collected over the whole of England and Wales. The average sample size for each year is 695.

The structure of the paper is as follows. We first describe the data from the FBS as well as our models and estimators for high yields in this dataset. We then give an account of the implementation of those techniques, first on the full dataset, then on the data stratified by location, and finally on the data stratified by spending on agricultural inputs, with an emphasis on estimated maximum yield levels. A Discussion section concludes with additional comments and ideas for further work.

## 2. Data and methods

### 2.1. Data: Farm Business Survey

The FBS collects information about farm businesses in England and Wales, to give a yearly overall perspective of the agricultural and economic performance of farms. Each year, approximately 2300 farms take part in the survey. On average, 695 were involved with the production of winter wheat from 2006 to 2015, each having 76 observed variables, among which were yield, region and fertilizer and crop protection costs, to which we restrict our attention. To take inflation into account, the financial data are adjusted to their 2010 equivalent [38]. Summary boxplots of the data are provided in figure 1. In terms of productivity, 2012 was a year of low wheat yields as a result of poor weather conditions, according to the UK Department for Environmental, Food and Rural Affairs (DEFRA) [39]. In 2015, DEFRA reported that wheat yields had reached their highest level since 1990 [40]; that year, the crops benefited from optimal growing conditions during the spring and summer months.

To remove any duplication of high-yielding farms, we decided that for each farm which took part in the survey at least once, we would retain its maximum yield over all the years in which they have contributed to the FBS. Constructing the yearly recordings in this manner results in a sample of $n = 1536$ unique farms. Theoretical and practical justification for this way of constructing the data can be found in the electronic supplementary material, Sections A and B. Furthermore, the farms incur a number of costs, among which are spending on agricultural inputs such as crop protection and fertilizers; these are used to evaluate the impact of the use of agricultural inputs on high yield levels. To ensure anonymity and guarantee reasonably large sample sizes, the farms' locations are studied using the macro-regions east England, north England, and west England and Wales. These regions are constructed by grouping together the NUTS1 administrative regions in the UK (figure 2) as follows:

— East England: East Midlands, East of England, London and South East England;
— North England: North East England, North West England and Yorkshire & Humberside;
— West England and Wales: West Midlands, South West England and Wales.

### 2.2. Method: extreme value analysis

Extreme value analysis provides a powerful statistical framework for the analysis of the highest values of a variable summarizing a physical or natural phenomenon [41,42], in our case yields. Denoting the yield of a farm by $X$, and given a high value $t$ of yield, extreme value theory proposes that a yield $X$ larger than $t$ approximately follows a generalized Pareto distribution. In other words, for high $t$,

$$\mathbb{P}(X - t \leq y \mid X > t) \approx H_{\gamma,\sigma(t)}(y) := 1 - \left(1 + \frac{\gamma y}{\sigma(t)}\right)^{-1/\gamma}, \tag{2.1}$$

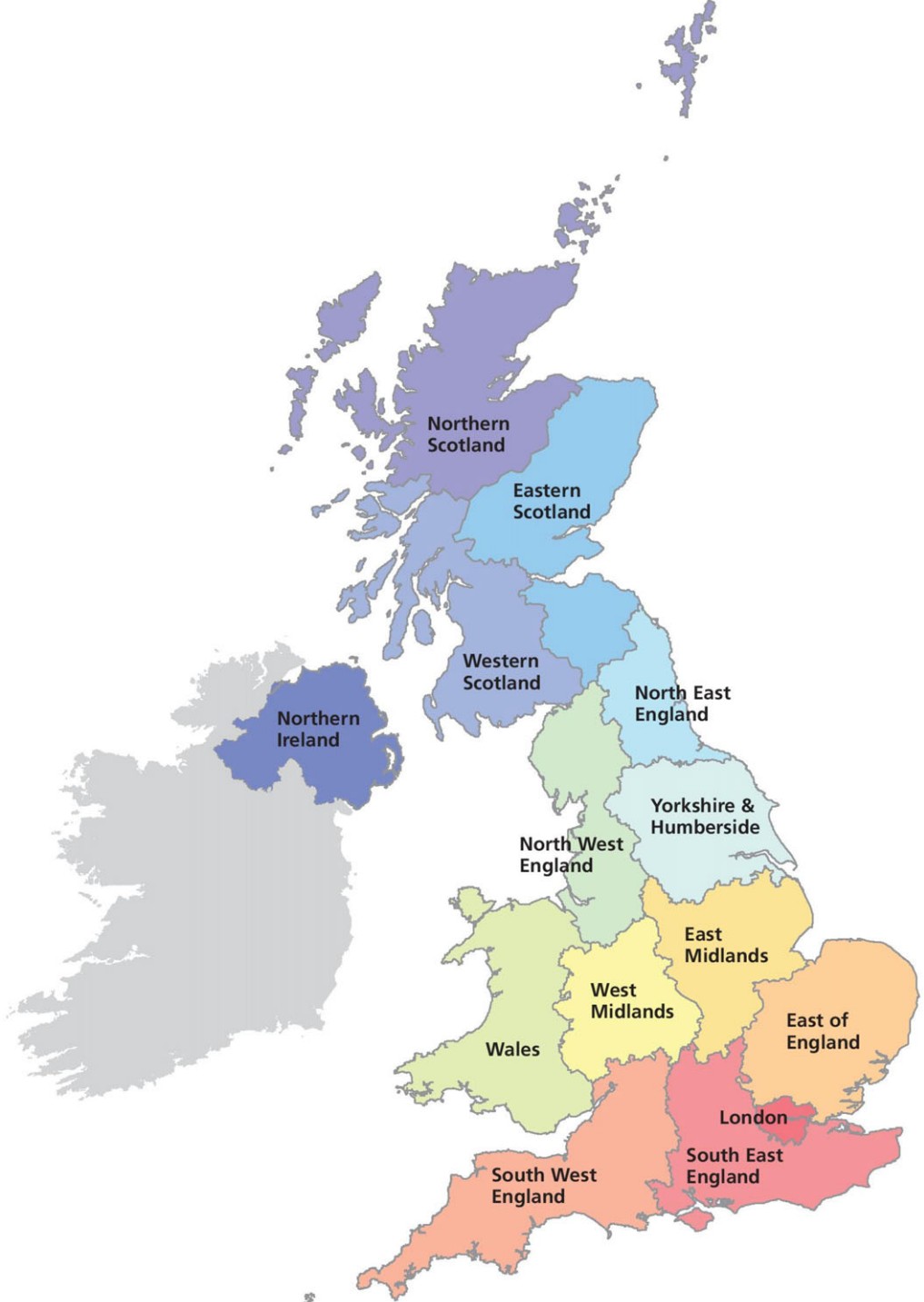

**Figure 2.** Administrative subdivision of the UK in NUTS1 regions (source: Met Office).

for all $y > 0$ such that $1 + \gamma y / \sigma(t) > 0$. In this equation, $\gamma$ is a shape parameter which controls the behaviour of the extremes of $X$, while $\sigma(t)$ is a positive scale parameter. For $\gamma = 0$, $H_{\gamma,\sigma(t)}(y)$ becomes $H_{\sigma(t)}(y) = 1 - \exp(-y/\sigma(t))$. A typical choice of the threshold $t$ is a high data point in the full sample, so that the final sample is made of the $k$ highest yields, where this effective sample size $k$ is small relative to the total size $n$ of the sample of data.

A popular method of parameter estimation for the shape and scale parameters in equation (2.1) based on collected data is maximum likelihood (ML), for which a full theoretical analysis is available [43]. In particular, when $\gamma > -1/2$ and under a classical second-order condition making it possible to quantify

the gap between the actual underlying right tail and the tail of the associated generalized Pareto distribution, the ML estimator $(\hat{\gamma}_k, \hat{\sigma}_k)$ of $(\gamma, \sigma_k)$, based on the $k$ highest yields within the sample of yield data, will be asymptotically normal:

$$\left(\hat{\gamma}_k, \frac{\hat{\sigma}_k}{\sigma_k}\right) \approx N_2\left((\gamma, 1), \frac{1}{k}\mathbf{V}\right), \quad \text{with } \mathbf{V} = \begin{pmatrix} (1+\gamma)^2 & -(1+\gamma) \\ -(1+\gamma) & 1+(1+\gamma)^2 \end{pmatrix}. \tag{2.2}$$

This allows approximate confidence intervals based on asymptotic normality for $(\gamma, \sigma_k)$ to be constructed.

In order to apply equation (2.2), a choice must be made for the parameter $k$. For small $k$, according to equation (2.2), the variance of $\hat{\gamma}_k$ will be very high, so that the curve $k \mapsto \hat{\gamma}_k$ will be very erratic; for large $k$, the estimation method will tend to use non-extreme yields and therefore introduce substantial bias in the results. Hence the desirability of choosing a value $k$ which is neither too small nor too large. In practice, such values of $k$ often appear in the form of a stability region where $k$ is high enough that the estimates $\hat{\gamma}_k$ have stabilized, but low enough that the extremes remain a recognizable feature of the data. Such techniques have been studied extensively in the probabilistic and statistical literature [44–48].

The type of extremes the data exhibits can then be determined based on the estimate of $\gamma$. In particular, $\gamma < 0$ implies that there exists a finite maximal value $x^*$ of the yield $X$: indeed, the condition $1 + \gamma y/\sigma(t) > 0$ in equation (2.1) entails in this case $y < -\sigma(t)/\gamma$. For such distributions, an estimate of the upper bound $x^*$ (also called *right endpoint*) is

$$\hat{x}_k^* = t_k - \frac{\hat{\sigma}_k}{\hat{\gamma}_k}, \tag{2.3}$$

where $t_k$ denotes the $(k+1)$th highest value in the sample. The uncertainty for this endpoint estimator can also be quantified, in the sense that, given that $\gamma < 0$ we have [42, §4.5.1 p. 147]

$$\hat{x}_k^* \approx x^* + \frac{1}{\sqrt{k}} \times \frac{\hat{\sigma}_k}{\hat{\gamma}_k^2} \times N(0, 1 + 4\gamma + 5\gamma^2 + 2\gamma^3 + 2\gamma^4). \tag{2.4}$$

Just as in equation (2.2), this approximation makes it possible to construct confidence intervals for the true maximum yield.

It is noteworthy that when the approximate confidence interval based on asymptotic normality for the shape parameter, produced via equation (2.2), contains 0, the confidence interval produced by this approximation may underestimate uncertainty at the upper end of the confidence interval, by not accounting for the theoretical possibility of a heavy tail which would imply an unbounded maximum yield. That being said, wheat production in the UK's temperate climate is characterized by high input–high output biological relationships, with farmers applying high input levels of nitrogen as they aim to produce for high yield [49] rather than economically optimal yields. It is also known that biological cropping systems, such as winter wheat farms, typically exhibit diminishing productivity functions with respect to input–output relationships [50,51], and even more strongly, that an over-application of inputs can lead to marginal yield reductions. In the context of the agricultural input-intensive UK commercial production of wheat, this means that the high levels of wheat yields that we observe in our data are quite likely to be of the order of magnitude of maximum wheat yield; in any event, it is very unlikely that an arbitrarily large yield is physically and biologically possible (a related point is made in [52]).

At the same time, the Gaussian confidence interval may overestimate uncertainty at the lower end of the confidence interval, since this lower bound is not constrained to be larger than the maximum value in the sample (a clear lower bound for the true value of the endpoint). This is especially important when the estimate $\hat{\gamma}_k$ is close to zero, because then the presence of the factor $1/\hat{\gamma}_k^2$ in equation (2.4) typically makes this lower bound unreasonably conservative. For this reason, we propose to use the interval

$$\left[\max\left(t_0, \hat{x}_k^* - \frac{1.96}{\sqrt{k}} \times \frac{\hat{\sigma}_k}{\hat{\gamma}_k^2} \times \sqrt{1 + 4\hat{\gamma}_k + 5\hat{\gamma}_k^2 + 2\hat{\gamma}_k^3 + 2\hat{\gamma}_k^4}\right),\right.$$
$$\left.\hat{x}_k^* + \frac{1.96}{\sqrt{k}} \times \frac{\hat{\sigma}_k}{\hat{\gamma}_k^2} \times \sqrt{1 + 4\hat{\gamma}_k + 5\hat{\gamma}_k^2 + 2\hat{\gamma}_k^3 + 2\hat{\gamma}_k^4}\right]$$

as an approximate 95% confidence interval for the maximum yield $x^*$, where $t_0$ denotes the maximum value in the sample. Note that truncating the interval at level $t_0$ does not affect its coverage probability in practice because, by definition, the true value $x^*$ of the right endpoint must be larger than $t_0$ with probability 1. An alternative technique that better accounts for the uncertainty at the

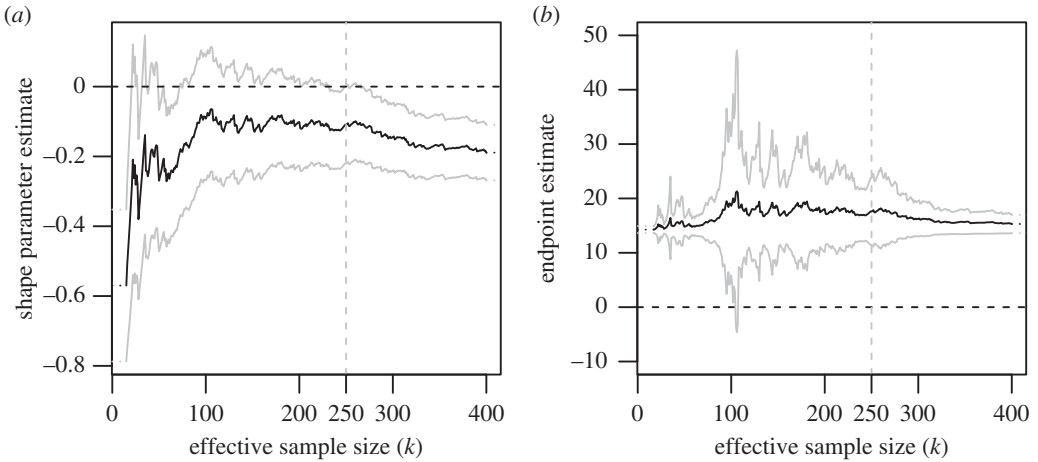

**Figure 3.** (a) Plot of the ML estimate of the shape parameter $\gamma$, (b) plot of the estimate of the endpoint $x^*$. Both plots give the estimates as a function of the effective sample size $k$ taken for the estimation, with corresponding approximate 95% Gaussian confidence intervals. Estimates for sample sizes smaller than 15 and greater than 400 are omitted due to large variance and large bias, respectively.

lower end of the confidence interval is given by the profile likelihood method for a once-in-$m$ years return level (e.g. [53]), letting $m \to \infty$. This method, however, will by construction produce a confidence interval unbounded to the right when the related confidence interval for the shape parameter contains 0, since it cannot exclude the possibility of a heavy tail and thus of an infinite right endpoint. In view of our above arguments on the impossibility of an arbitrarily large yield, we do not think that using such intervals is advisable in our context and would therefore recommend to adopt the above calculation based on the Gaussian approximation; we have nonetheless included some profile likelihood calculations of extreme return levels for yield in the electronic supplementary material, Section C to give an idea of how this method behaves on the present dataset.

Estimating the maximum yield under current farming conditions and providing such confidence intervals for these estimates is precisely the objective of our data analysis in the next section, carried out on the FBS data.

# 3. Results

To estimate the maximum value of yield, we first have to choose the threshold for our extreme value modelling of yield, or equivalently the number $k$ of high data points employed. We do so by representing the curve of ML estimates of the shape parameter $\gamma$ as a function of $k$ in figure 3. These estimates are calculated using the shape function of the R package evir [54]. The R codes used to produce the analyses of the data are provided in the electronic supplementary material. Figure 3 suggests that the ML estimate $\widehat{\gamma}_k$ is very stable for $k$ between 100 and 250, implying that the largest 250 observed yields are a suitable sample of data on which to base our analysis of high yields. The choice $k = 250$ corresponds to taking the threshold $t = t_{250} = 10.69$ tonnes per hectare, and the ML estimate for the shape parameter $\gamma$ is then $\widehat{\gamma}_{250} = -0.11\,(-0.22, 0.00)$ (throughout, all confidence intervals are calculated at the approximate 95% confidence level). With this negative shape parameter estimate and the corresponding ML estimate $\widehat{\sigma}_{250} = 0.76\,(0.65, 0.91)$ for the scale parameter (produced using the gpd function part of the R package evir), we find, using formula (2.3), a finite right endpoint estimate $\hat{x}^*_{250} = t_{250} - \hat{\sigma}_{250}/\hat{\gamma}_{250} = 17.60\,(14.02, 23.75)$ tonnes per hectare. These results, along with those of our subsequent analyses, are shown in table 1. This estimate of a finite upper bound for winter wheat yield agrees with the physical intuition that yield per hectare should be bounded by a maximum yield which cannot be exceeded. The current verified records for UK and worldwide wheat yields are 16.52 (observed in 2015) and 16.79 (observed in New Zealand in 2017 and confirmed by Guinness World Records) tonnes per hectare, suggesting that our estimated value of 17.60 tonnes per hectare is a sensible estimate of this maximum possible yield.

Although this extreme value analysis of winter wheat yield provides an estimate of the maximum attainable yield per hectare, this does not give any idea of the potential variation of wheat yields

**Table 1.** Maximum yield level estimates $\hat{x}^*$ for the full dataset and the data stratified with respect to region or spending on agricultural inputs, along with a summary of sample sizes, threshold choices, shape estimates $\hat{\gamma}$ and scale estimates $\hat{\sigma}$. Numbers in brackets next to shape, scale and maximum yield estimates represent approximate 95% confidence intervals.

| variable | $n$ | $k$ | $t$ | shape estimate $\hat{\gamma}$ | scale estimate $\hat{\sigma}$ | $\hat{x}^* = t - \hat{\sigma}/\hat{\gamma}$ |
|---|---|---|---|---|---|---|
| yield | 1536 | 250 | 10.69 | − 0.11 | 0.76 | 17.60 |
| | | | | (− 0.22, 0.00) | (0.65, 0.91) | (14.02, 23.75) |
| *location* | | | | | | |
| West England and Wales | 435 | 115 | 9.76 | − 0.10 | 0.80 | 17.68 |
| | | | | (− 0.27, 0.06) | (0.65, 1.07) | (13.25, 29.11) |
| North England | 331 | 68 | 10.58 | − 0.16 | 0.87 | 15.91 |
| | | | | (− 0.36, 0.03) | (0.67, 1.27) | (13.59, 21.20) |
| East England | 770 | 125 | 10.84 | − 0.11 | 0.74 | 17.81 |
| | | | | (− 0.26, 0.05) | (0.60, 0.96) | (14.02, 26.98) |
| *inputs* | | | | | | |
| low (input $<$ 271.5) | 512 | 90 | 9.93 | − 0.23 | 0.99 | 14.27 |
| | | | | (− 0.39, − 0.07) | (0.79, 1.34) | (12.85, 16.52) |
| medium (271.5 $\leq$ input $<$ 370.1) | 512 | 80 | 10.67 | − 0.11 | 0.65 | 16.40 |
| | | | | (− 0.31, 0.08) | (0.50, 0.92) | (13.28, 24.99) |
| high (input $>$ 370.1) | 512 | 100 | 10.96 | − 0.09 | 0.75 | 19.18 |
| | | | | (− 0.27, 0.09) | (0.60, 1.03) | (14.02, 33.58) |

depending on geography or growing conditions. These two questions are the focus of our next two refined analyses.

## 3.1. Difference in geographical regions

Because there is variation in winter wheat yields across England and Wales [40], it is important to carry out regional analyses of yield. Past studies have, for instance, assessed the evolution over time of winter wheat yields for 13 administrative regions in the UK [34]. It is, however, likely that dividing our sample of $n = 1536$ data points according to such a fine regional partition will result in samples that are too small to be able to carry out a meaningful extreme value analysis. To address this need for reasonable sample sizes and our idea of identifying potential regional variation of high yields, we decided to regroup farms using the macro-regions west England and Wales, north England and east England. This results in sample sizes of, respectively, 435, 331 and 770, which are appropriately large for our extreme value analysis. We also note that, in addition to containing the highest number of farms, east England has a larger average yield per hectare figure compared to the other two regions. Based on this geographical subdivision, we carry out an extreme value analysis similar to the global analysis of the previous section to model regional high yields. This is justified by classical likelihood ratio tests [55,56] based on the generalized Pareto model, which show that the model appropriate to the description of high yields depends indeed on the chosen region; we do not report the results of such tests here for the sake of brevity. The regional shape parameter estimates, as a function of effective sample size, are plotted in figure 4.

As table 1 shows, all three regions reassuringly give negative shape parameter estimates, albeit with wider confidence intervals; this was expected since stratifying decreases the available sample size and therefore increases uncertainty. These shape parameter estimates, together with matching estimates of the regional scale parameter, make it possible to produce estimates of regional upper bounds for yield using formula (2.3). These estimates are 17.68 (13.25, 29.11) tonnes per hectare for west England and Wales, 15.91 (13.59, 21.20) for north England, and 17.81 (14.02, 26.98) for east England. The wide confidence intervals on these extreme value estimates make it impossible to suggest that, at the 95% level, there are

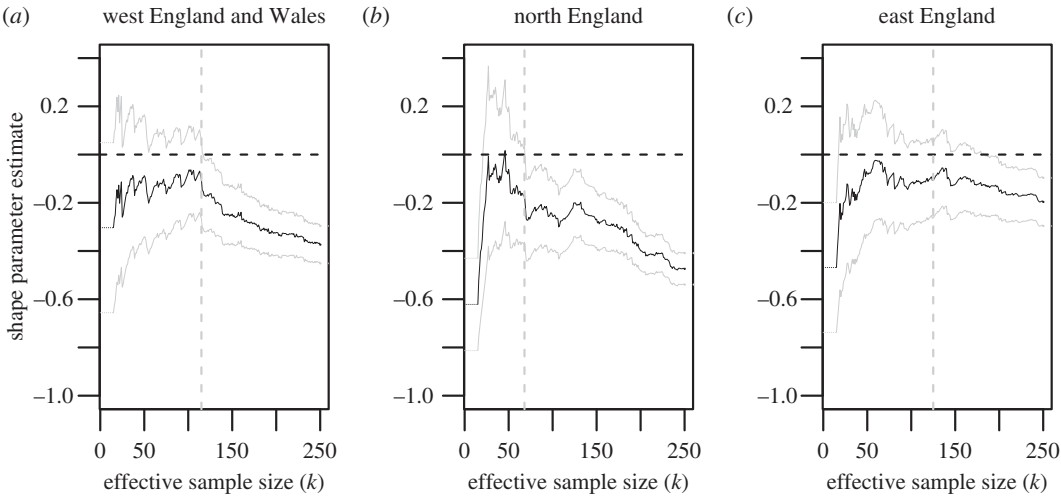

**Figure 4.** ML estimates of $\gamma$, for west England and Wales (*a*), north England (*b*) and east England (*c*).

regional differences between maximal yields across the three considered regions, although we do mention that the point estimate of maximal yield is noticeably lower for north England. We conclude this analysis by mentioning that although the point estimates of maximal yield in west England and Wales and east England are slightly higher than the point estimate across the whole dataset, the increase is statistically insignificant and appears to be due to the fluctuations of the maximum yield estimate as a function of the effective sample size $k$. There is therefore no inconsistency between these stratified results and our earlier global analysis.

## 3.2. Difference in inputs

Fertilizer and crop protection use for large-scale agricultural activities has long been at the centre of a vigorous debate. A number of academic studies across disciplines have debated their effects on public health and the environment along with how to effectively control their use [57–63]; outside academic contexts, European Union policymakers and legislators voted in April 2018 an almost complete ban on neonicotinoids due to their effects on honeybees and other pollinators. This motivates our idea of assessing whether the effect of agricultural inputs on maximal wheat yield levels can be identified. We divide the sample of $n = 1536$ farms into three equally sized groups according to their expenditure on fertilizers and crop protection: low (less than £271.50 per hectare per year, corresponding to the bottom third in terms of expenditure), medium (between £271.50 and £370.10 per hectare per year, corresponding to the middle third), and high (greater than £370.10 per hectare per year, corresponding to the top third). Based on this stratification by spending, and again in view of likelihood ratio tests indicating to us that the appropriate model for high yields indeed depends on input level, we carry out an extreme value analysis similar to the above regional analysis. Shape parameter estimates are represented in figure 5.

All three categories give negative shape parameter estimates, although the estimate for low input levels lies outside the confidence interval for the estimate of the shape parameter estimate of the full yield data, suggesting a significant difference in the behaviour of high yields for low spenders. The associated upper limit estimates for yield are 14.27 (12.85, 16.52), 16.40 (13.28, 24.99) and 19.18 (14.02, 33.58) for low, medium and high use of inputs, respectively. The value and uncertainty on the maximal yield estimates for low spending on inputs do suggest that the use of fertilizer and crop protection improves the maximum attainable yield; however, and despite a point estimate of maximal yield being higher for the biggest consumers of these inputs than for average users, the uncertainty on our estimates does not provide significant evidence that spending a larger amount of capital on fertilizer and crop protection improves maximal yield levels.

## 4. Discussion and conclusion

Our analysis of 10 years of recent winter wheat production data, collected in England and Wales by the FBS, indicates that annual winter wheat yields per hectare have a finite upper bound which we estimate

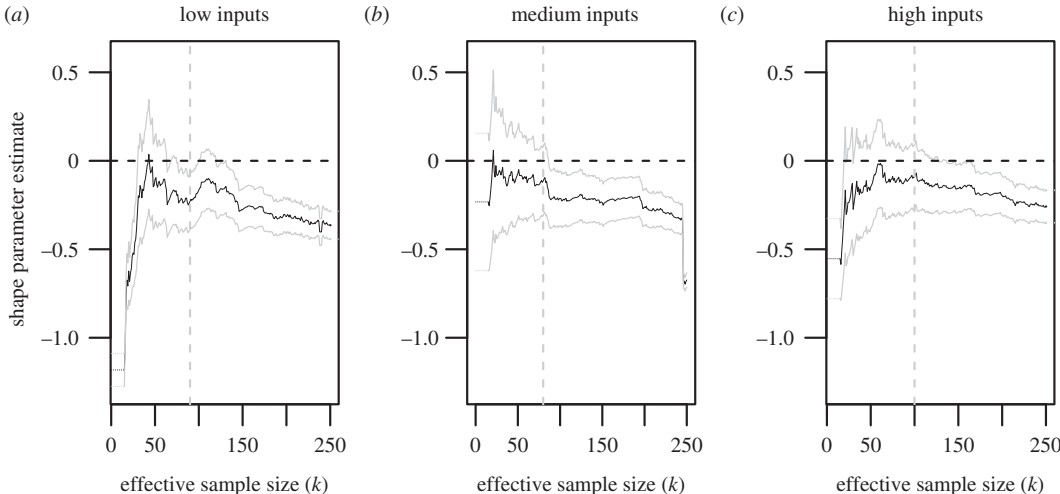

**Figure 5.** ML estimates of $\gamma$, for low input levels (*a*), medium input levels (*b*) and high input levels (*c*).

to be 17.60 tonnes. Our model, based on the use of a generalized Pareto distribution suggested by the framework of extreme value analysis, was also adapted to the estimation of regional maximal yields and maximal yields as a function of spending on agricultural inputs. These estimates seem plausible, and show that:

— Although the maximum yield point estimate for north England is lower than the corresponding ones for west England and Wales and east England, there is insufficient statistical evidence to suggest that north England farms cannot reach the estimated maximum yield of 17.60 tonnes per hectare.
— There is an increase in maximum yield from low to high use of fertilizer and crop protection, although the difference between the maximal yields of medium and high spenders on these inputs is not statistically significant.

To use our ML estimators of the shape and scale parameters, and then deduce an estimate of the right endpoint of yield, we had to make the distributional assumption that yields above a sufficiently high threshold approximately follow a generalized Pareto distribution. The quality of this approximation is a critical factor in the performance of the estimators, and may lead to poor estimates if the underlying distribution of high yields is too far from our model. To make our estimates robust against a potential departure from the model, we could have presented a semi-parametric approach instead, such as probability weighted moment estimators [64] or the moment estimator [65]. Both of these estimators are flexible in the sense that their validity is not rooted in the generalized Pareto assumption, but the price to pay for this is their higher asymptotic variance compared to the ML estimator [42]. It turns out that, in our preliminary analyses, these semi-parametric alternatives gave shape and scale parameter estimates close to the ML estimate, which encouraged us to prefer the latter for its narrower confidence intervals.

The second part of our analysis was an effort to assess the dependence of the maximum yield on location of a farm. The point estimate of maximal yield in north England, which is 15.91 tonnes per hectare, is actually lower than the verified record for this region, which is also the UK record of 16.52 tonnes per hectare, attained in a Northumberland farm in 2015. This data point, which is not part of the data from the FBS and hence not taken into account in our methodology, is well within the confidence interval calculated for the maximum yield in north England and thus not inconsistent with our results. Analysing the reasons behind this extremely high yield reveals that, while the north of England typically suffers from increased rainfall, lower temperatures and limited sunshine compared to the southern part of the UK, this was not the case in 2015 [40]. A fruitful avenue of further work would be to gather sufficient climate data in order to design a model of the influence of weather parameters upon winter wheat yields. Such a model would also be very useful when accounting for the effect of climate change on maximum yield levels.

The third and final part of our extreme value analysis, stratified with respect to spending on agricultural inputs, suggested that there is not a statistically established increase in maximum yield

arising from a large use of crop protection and fertilizers. Our findings, consistent with previous studies [49,66], indicate the potential for an upper-level marginal input use reduction while still obtaining high yields, providing high food production potential, increased farmer profit and reduced environmental footprint. Our statistical analyses demonstrate no significant difference in extreme yield between medium and high input use, and that additionally there was no significant difference in maximum yield across the three regions within the dataset, implying that soil type and weather variation are, on aggregate, not the main drivers of high yields within the data.

One important question which we have not addressed here is to determine what factors make the large difference between a yield close to maximum (estimated to be 17.60 tonnes per hectare) and typical yield (approx. 8 tonnes per hectare). Attention to detail in agricultural production practice has been previously cited as a key profitability driver [67], and exploring the managerial drivers of performance with an extreme value theory approach represents a potentially fruitful area of research work. It would also be informative to re-test the hypothesis of the difference in maximum attainable yields against different fertilizer and crop protection input use levels from a larger sample of data, for example, drawn from European wide data or from the USA. This would reduce the width of the confidence intervals for the estimates of maximum yield stratified according to spending in agricultural inputs. The potentially large yield gains to be made, starting from average yield levels, imply that detailed farm-level studies of agricultural practice with statistically relevant numbers of observations would be worthwhile. Another interesting question, which is beyond the scope of the present paper and to be addressed in future research, is to find a precise model for the description of high levels of yield as a function of agricultural input use and location. This could be done by, for instance, letting the scale or shape parameter (or both) vary smoothly as a function of input level or geographical coordinates, as described for instance in [68]. Such an analysis would allow for the prediction of the high and maximum levels of yield attainable under given biological and physical circumstances, and would thus be important for agricultural policy and decision-making.

Data accessibility. The FBS data (2006–2015) is under the control of the UK Data Service. The raw data are available upon satisfactory completion of a Special Licence request with the UK Data Service at www.ukdataservice.ac.uk, under the references SN 5838, SN 6144, SN 6387, SN 6682, SN 6967, SN 7231, SN 7461, SN 7659, SN 7914 and SN 8158. The economic data used to adjust monetary recordings to their 2010 equivalent is available in the electronic supplementary material and at www.gov.uk/government/statistics/agricultural-price-indices.
Authors' contributions. P.W. undertook initial data collation and manipulation. E.G.M. and G.S. undertook the analyses and prepared the draft manuscript. E.G.M., G.S. and A.T.A.W. developed the statistical interpretation. P.W. and N.M.J.C. provided further background and interpretation within the agricultural context. All authors conceived the study and reviewed the manuscript.
Competing interests. The authors declare no competing interests.
Funding. E.G.M.'s work has been funded under the Leverhulme Doctoral Scholarship grant scheme as part of the Modelling and Analytics for a Sustainable Society PhD programme at the University of Nottingham.
Acknowledgements. The authors would like to thank the referees and the Associate Editor for their valuable suggestions which have led to a substantially improved version of this paper.

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
