## [Reviewer comments · Royal Society Open Science]

Review History

RSOS-190242.R0 (Original submission)

Review form: Reviewer 1

Is the manuscript scientifically sound in its present form?

Yes

Are the interpretations and conclusions justified by the results?

Yes

Is the language acceptable?

Yes

Is it clear how to access all supporting data?

Yes

Do you have any ethical concerns with this paper?

No

Have you any concerns about statistical analyses in this paper?

No

Recommendation?

Reject

Comments to the Author(s)

Please see attached report (Appendix A).

Review form: Reviewer 2**Is the manuscript scientifically sound in its present form?**

Yes

Are the interpretations and conclusions justified by the results?

Yes

Is the language acceptable?

Yes

Is it clear how to access all supporting data?

Yes

Do you have any ethical concerns with this paper?

No

Have you any concerns about statistical analyses in this paper?

Yes

Recommendation?

Major revision is needed (please make suggestions in comments)

Comments to the Author(s)

Overall, I found this to be a clear, concise and very well written paper, that provides an novel, interesting and practically relevant application of a relatively sophisticated form of statistical modelling. The paper has a clear application, the exposition of the methodology is good, and the level of technical detail seemed to me to be appropriate. I thought that the use of Extreme Value Theory to address this problem was defensible, worthwhile and interesting. The discussion of the outputs, in relation to agricultural production, seemed very clear, and I thought that this constituted an accurate reflection of the findings of the analysis. The text also, I felt, properly acknowledged the caveats and limitations associated with the approach used.

I feel, however, that one major revision, and a few minor revisions, are needed in order to ensure that the selection of methods and the interpretation of the results are fully defensible.

PROPOSED MAJOR REVISION

A. In situations where a farm has data over multiple years, the authors have selected the maximum yield for that form, across all years, to go into the analysis. Does this approach not create potential for bias, since the maximum yield per farm will tend to be positively related to the number of years used in constructing the maximum value? (i.e., all else being equal, the maximum observed yield for a farm with 5 years of data will tend on average to be systematically larger than the maximum yield for a farm with two years of data...). If that is the case, I think some evaluation of the likely consequences of this is needed (e.g. by adapting the model to account for differences in the number of years, and/or by assessing whether the results change

qualitatively when an alternative approach to data selection which avoids this issue - such as randomly selecting a single year of data for each farm - is used).

PROPOSED MINOR REVISIONS

1. I was rather surprised that in many cases (if I am reading it correctly) the lower limits of the 95% confidence intervals of the estimate for the upper endpoint were lower than the maximum observed value in the data used to derive these estimates, and in some cases seemed much lower. This seems counter-intuitive, given that the upper endpoint must presumably, in reality, be no lower than the maximum observed value. Why does this occur, and does the fact that it occurs indicate an issue with the model/inference? I think the authors need to comment on this, and, if they are convinced that this is not an issue, provide a justification for that.

(More specifically, does this phenomenon arise from the normality assumption in Equations 2 and 4, and if so could it indicate an issue with that assumption? [e.g. the estimator is presumably only asymptotically normal, so could that assumption be breaking down substantially for samples of this size?]. Is there any possibility to avoid this issue by using an alternative approach to construction of the confidence interval - e.g. profile likelihood, as I think is used in Coles (2001) and the associated "ismev" R package, which I believe avoids an assumption of symmetry.)

2. Would it be possible to formally test whether the differences between regions, and levels of input, are significant? - e.g. using a likelihood ratio test? It seems slightly unsatisfactory that at the moment the only way to evaluate whether the differences are significant are not is to look at whether the confidence intervals for groups overlap each other (which is not, I think, equivalent to a formal test of a significant difference between regions/input levels).

3. The statement "maximum yield levels appear to plateau as a function of crop protection and fertiliser use" is made in the abstract, but I couldn't follow the rationale for this conclusion - the difference between the point estimate of the upper endpoint of yield under the "high inputs" category and "medium inputs category" seems to be larger than the difference between the endpoints under the "medium input" and "low input" categories, so I couldn't see that the results of the analysis provided evidence for a plateau? (I certainly agree that the results could be consistent with the existence of a plateau, but I don't quite follow the rationale for how they provide positive evidence for the existence of one, especially since the uncertainty associated with the estimated upper endpoint for each input level is very large).

4. "does not suggest that spending a larger amount of capital on fertiliser and crop protection significantly improves": I think the wording here rather confuses statistical significance and agricultural significance. I would suggest that rephrasing this to "does not provide significant evidence that spending a larger amount of capital on fertiliser and crop protection improves" provides a more accurate wording.

5. Is Equation 4 conditional on the shape parameter being negative? If so, will it tend to underestimate uncertainty in situations where the confidence interval for the shape parameter overlaps zero? If that is the case, I think that this is not necessarily a problem (especially since in this application the confidence intervals rarely do overlap zero) but that it should be explicitly noted in the text as a caveat associated with these confidence intervals.

Review form: Reviewer 3

Is the manuscript scientifically sound in its present form?

Yes

Are the interpretations and conclusions justified by the results?

No

Is the language acceptable?

Yes

Is it clear how to access all supporting data?

Yes

Do you have any ethical concerns with this paper?

No

Have you any concerns about statistical analyses in this paper?

Yes

Recommendation?

Reject

Comments to the Author(s)

My comments are included in report (Appendix B).

Decision letter (RSOS-190242.R0)

23-May-2019

Dear Professor Mitchell:

Manuscript ID RSOS-190242 entitled "Operating at the extreme: estimating the upper yield boundary of winter wheat production in commercial practice" which you submitted to Royal Society Open Science, has been reviewed. The comments from reviewers are included at the bottom of this letter.

In view of the criticisms of the reviewers, the manuscript has been rejected in its current form. However, a new manuscript may be submitted which takes into consideration these comments.

Please note that resubmitting your manuscript does not guarantee eventual acceptance, and that your resubmission will be subject to peer review before a decision is made.

Your resubmitted manuscript should be submitted by 20-Nov-2019. If you are unable to submit by this date please contact the Editorial Office.

Kind regards,
Andrew Dunn

Royal Society Open Science Editorial Office
Royal Society Open Science
openscience@royalsociety.org

on behalf of Professor Ruth King (Associate Editor) and Mark Chaplain (Subject Editor)
openscience@royalsociety.org

Associate Editor Comments to Author (Professor Ruth King):

The manuscript has been reviewed by three referees and myself. The paper overall is very well written with a clear motivation and application to real data. However, a number of significant issues have been identified within the analysis that raise particular concerns. The primary issues raised relate to:

- modelling considerations - as is the case with any applied problem there will be multiple ways the problem can be approached - and everyone will have a slightly different ideas. However - in this case the referees do provide some useful suggestions regarding the model which I think could significantly improve some of the aspects of the paper - particularly in relation to the spatial aspect and assessment of differences between the regions in a more formal manner particularly given that this is one of the focuses of the paper.

- the exact extracted data that is fitted within the analysis could have additional variability due to the length of time the farms are in the study. This should be considered further and the robustness of this investigated.

- the (approximate) confidence intervals that are used lead to contradictory intervals (in terms of lower limits being below the observed data) - these should be investigated further and the referees provide a useful suggestion here with regard to alternative approaches, particularly in relation to a profile likelihood approach.

Associate Editor: 2
Comments to the Author:
(There are no comments.)

Reviewers' Comments to Author:
Reviewer: 1

Comments to the Author(s)
Please see attached report.

Reviewer: 2

Comments to the Author(s)

Overall, I found this to be a clear, concise and very well written paper, that provides an novel, interesting and practically relevant application of a relatively sophisticated form of statistical modelling. The paper has a clear application, the exposition of the methodology is good, and the level of technical detail seemed to me to be appropriate. I thought that the use of Extreme Value Theory to address this problem was defensible, worthwhile and interesting. The discussion of the outputs, in relation to agricultural production, seemed very clear, and I thought that this constituted an accurate reflection of the findings of the analysis. The text also, I felt, properly acknowledged the caveats and limitations associated with the approach used.

I feel, however, that one major revision, and a few minor revisions, are needed in order to ensure that the selection of methods and the interpretation of the results are fully defensible.

PROPOSED MAJOR REVISION

A. In situations where a farm has data over multiple years, the authors have selected the maximum yield for that farm, across all years, to go into the analysis. Does this approach not create potential for bias, since the maximum yield per farm will tend to be positively related to the number of years used in constructing the maximum value? (i.e., all else being equal, the maximum observed yield for a farm with 5 years of data will tend on average to be systematically larger than the maximum yield for a farm with two years of data...). If that is the case, I think some evaluation of the likely consequences of this is needed (e.g. by adapting the model to account for differences in the number of years, and/or by assessing whether the results change qualitatively when an alternative approach to data selection which avoids this issue - such as randomly selecting a single year of data for each farm - is used).

PROPOSED MINOR REVISIONS

1. I was rather surprised that in many cases (if I am reading it correctly) the lower limits of the 95% confidence intervals of the estimate for the upper endpoint were lower than the maximum observed value in the data used to derive these estimates, and in some cases seemed much lower. This seems counter-intuitive, given that the upper endpoint must presumably, in reality, be no lower than the maximum observed value. Why does this occur, and does the fact that it occurs indicate an issue with the model/inference? I think the authors need to comment on this, and, if they are convinced that this is not an issue, provide a justification for that.

(More specifically, does this phenomenon arise from the normality assumption in Equations 2 and 4, and if so could it indicate an issue with that assumption? [e.g. the estimator is presumably only asymptotically normal, so could that assumption be breaking down substantially for samples of this size?]. Is there any possibility to avoid this issue by using an alternative approach to construction of the confidence interval - e.g. profile likelihood, as I think is used in Coles (2001) and the associated "ismev" R package, which I believe avoids an assumption of symmetry.)

2. Would it be possible to formally test whether the differences between regions, and levels of input, are significant? - e.g. using a likelihood ratio test? It seems slightly unsatisfactory that at the moment the only way to evaluate whether the differences are significant are not is to look at whether the confidence intervals for groups overlap each other (which is not, I think, equivalent to a formal test of a significant difference between regions/input levels).

3. The statement "maximum yield levels appear to plateau as a function of crop protection and fertiliser use" is made in the abstract, but I couldn't follow the rationale for this conclusion - the difference between the point estimate of the upper endpoint of yield under the "high inputs" category and "medium inputs category" seems to be larger than the difference between the endpoints under the "medium input" and "low input" categories, so I couldn't see that the results of the analysis provided evidence for a plateau? (I certainly agree that the results could be consistent with the existence of a plateau, but I don't quite follow the rationale for how they provide positive evidence for the existence of one, especially since the uncertainty associated with the estimated upper endpoint for each input level is very large).

4. "does not suggest that spending a larger amount of capital on fertiliser and crop protection significantly improves": I think the wording here rather confuses statistical significance and agricultural significance. I would suggest that rephrasing this to "does not provide significant evidence that spending a larger amount of capital on fertiliser and crop protection improves" provides a more accurate wording.

5. Is Equation 4 conditional on the shape parameter being negative? If so, will it tend to underestimate uncertainty in situations where the confidence interval for the shape parameter overlaps zero? If that is the case, I think that this is not necessarily a problem (especially since in

this application the confidence intervals rarely do overlap zero) but that it should be explicitly noted in the text as a caveat associated with these confidence intervals.

Reviewer: 3

Comments to the Author(s)

My comments are included in report

Author's Response to Decision Letter for (RSOS-190242.R0)

See Appendix C.

RSOS-191226.R0

Review form: Reviewer 1

Is the manuscript scientifically sound in its present form?

Yes

Are the interpretations and conclusions justified by the results?

Yes

Is the language acceptable?

Yes

Do you have any ethical concerns with this paper?

No

Have you any concerns about statistical analyses in this paper?

No

Recommendation?

Accept as is

Comments to the Author(s)

No further comments: all of my previous remarks have been answered in this revision.

Review form: Reviewer 2

Is the manuscript scientifically sound in its present form?

Yes

Are the interpretations and conclusions justified by the results?

Yes

Is the language acceptable?

Yes

Do you have any ethical concerns with this paper?

No

Have you any concerns about statistical analyses in this paper?

Yes

Recommendation?

Accept with minor revision (please list in comments)

Comments to the Author(s)

The authors have dealt very satisfactorily with all of the minor revisions that I raised in my previous review.

In relation to the one major revision that I proposed, which was concerned with the impact of variations in sample size between farms, the authors presented a clear argument in their response for why they do not regard this as being a problematic issue. The technical details of their argument seem sound, but I'm a bit concerned about the plausability of the assumptions underpinning it - if I have understood correctly, the argument relies on the assumption that annual yields for individual farms are i.i.d. (independent and identically distributed)? In practice, though, wouldn't we expect correlation between years within a farm, and between farms within a year, with the result that an analysis of individual annual farm yields would conventionally contain random effects for both "year" and "farm" - i.e. would not assume that annual farm-level yields were i.i.d.?

Does the technical argument for why variations in sample size do not alter the underlying distribution of maxima still hold if the i.i.d. assumption is violated, or is the contention of the authors that the i.i.d. assumption is reasonable in that context of these data? In either case, I think some justification for this is needed.

Review form: Reviewer 3

Is the manuscript scientifically sound in its present form?

No

Are the interpretations and conclusions justified by the results?

No

Is the language acceptable?

Yes

Do you have any ethical concerns with this paper?

No

Have you any concerns about statistical analyses in this paper?

Yes

Recommendation?

Reject

Comments to the Author(s)

See report (Appendix D).

Decision letter (RSOS-191226.R0)

02-Oct-2019

Dear Professor Mitchell:

Manuscript ID RSOS-191226 entitled "Operating at the extreme: estimating the upper yield boundary of winter wheat production in commercial practice" which you submitted to Royal Society Open Science, has been reviewed. The comments from reviewer(s) are included at the bottom of this letter.

In view of the criticisms of the reviewer(s), we must again decline the manuscript for publication in Royal Society Open Science at this time. However, a new manuscript may be submitted which takes into consideration these additional comments and concerns.

Please note that resubmitting your manuscript does not guarantee eventual acceptance, and that your resubmission will be subject to re-review by the reviewer(s) before a decision is rendered.

You will be unable to make your revisions on the originally submitted version of your manuscript. Instead, revise your manuscript using a word processing program and save it on your computer.

You may also click the below link to start the resubmission process (or continue the process if you have already started your resubmission) for your manuscript. If you use the below link you will not be required to login to ScholarOne Manuscripts.

*** PLEASE NOTE: This is a two-step process. After clicking on the link, you will be directed to a webpage to confirm. ***

https://mc.manuscriptcentral.com/rsos?URL_MASK=c8d8dbb201c1419d936672aa02b4f18b

Because we are trying to facilitate timely publication of manuscripts submitted to Royal Society Open Science, your resubmitted manuscript should be submitted by 31-Mar-2020. If you are unable to submit by this date please contact the Editorial Office for options.

We look forward to a resubmission.
Best regards,

Lianne Parkhouse
Royal Society Open Science
openscience@royalsociety.org

on behalf of Professor Ruth King (Associate Editor) and Mark Chaplain (Subject Editor)
openscience@royalsociety.org

Associate Editor Comments to Author (Professor Ruth King):

The revised manuscript has been reviewed by the same three reviewers as the initial submission. However, there are still significant concerns regarding the revised version of the manuscript, particularly from reviewers 2 and 3, who both feel that their previous comments have not been

satisfactorily addressed. For example, this includes, in particular, the construction of the confidence intervals (and associated issues with different approaches), as well as the argument associated with the variations in sample size, and the associated presentation of return levels.

Reviewer comments to Author:

Reviewer: 1

Comments to the Author(s)

No further comments: all of my previous remarks have been answered in this revision.

Reviewer: 2

Comments to the Author(s)

The authors have dealt very satisfactorily with all of the minor revisions that I raised in my previous review.

In relation to the one major revision that I proposed, which was concerned with the impact of variations in sample size between farms, the authors presented a clear argument in their response for why they do not regard this as being a problematic issue. The technical details of their argument seem sound, but I'm a bit concerned about the plausability of the assumptions underpinning it - if I have understood correctly, the argument relies on the assumption that annual yields for individual farms are i.i.d. (independent and identically distributed)? In practice, though, wouldn't we expect correlation between years within a farm, and between farms within a year, with the result that an analysis of individual annual farm yields would conventionally contain random effects for both "year" and "farm" - i.e. would not assume that annual farm-level yields were i.i.d.?

Does the technical argument for why variations in sample size do not alter the underlying distribution of maxima still hold if the i.i.d. assumption is violated, or is the contention of the authors that the i.i.d. assumption is reasonable in that context of these data? In either case, I think some justification for this is needed.

Reviewer: 3

Comments to the Author(s)

See report

Author's Response to Decision Letter for (RSOS-191226.R0)

See Appendix E.

RSOS-191919.R0

Review form: Reviewer 2

Is the manuscript scientifically sound in its present form?

Yes

Are the interpretations and conclusions justified by the results?

Yes

Is the language acceptable?

Yes

Do you have any ethical concerns with this paper?

No

Have you any concerns about statistical analyses in this paper?

No

Recommendation?

Accept as is

Comments to the Author(s)

I feel that the authors have now provided a comprehensive and convincing response to the remaining issue that I had raised in my previous review, so I do not believe that any further changes or additions are required

Review form: Reviewer 3

Is the manuscript scientifically sound in its present form?

Yes

Are the interpretations and conclusions justified by the results?

Yes

Is the language acceptable?

Yes

Do you have any ethical concerns with this paper?

No

Have you any concerns about statistical analyses in this paper?

No

Recommendation?

Accept as is

Comments to the Author(s)

The main dispute related to the use of corrected versions of asymptotic confidence interval based on normality and those from profile likelihood methods is addressed in the revised version.

The uncertainty surrounding estimating the maximum value of a given process still appears to me to be tremendous, but if necessarily needed to be estimated, then I find the methods presented in the current version useful. Alternative and useful ways might also be to report or support decisions based on return levels with finite return periods.

The new edits and changes are accurate and I am happy to recommend the paper to the journal.

Decision letter (RSOS-191919.R0)

18-Mar-2020

Dear Professor Mitchell,

I am pleased to inform you that your manuscript entitled "Operating at the extreme: estimating the upper yield boundary of winter wheat production in commercial practice" is now accepted for publication in Royal Society Open Science.

Royal Society Open Science operates under a continuous publication model. Your article will be published as soon as it is ready for publication, and this will be the final version of the paper. As such, it can be cited immediately by other researchers. As the issue version of your paper will be the only version to be published I would advise you to check your proofs thoroughly as changes cannot be made once the paper is published.

Articles are normally press released. For this to be effective we set an embargo on news coverage corresponding to the publication date of the article. We request that news media and the authors do not publish stories ahead of this embargo (when final version of the article is available). Please see the Royal Society Publishing guidance on how you may share your accepted author manuscript at <https://royalsociety.org/journals/ethics-policies/media-embargo/>.

on behalf of Professor Ruth King (Associate Editor) and Mark Chaplain (Subject Editor)
openscience@royalsociety.org

Associate Editor Comments to Author (Professor Ruth King):

Comments to the Author:

Apologies for the delay in this paper - and many thanks for your patience. Both myself and the outstanding reviewers appreciate the effort that you have gone to in order to address concerns raised - and the paper will now make a nice contribution to the journal.

Reviewer comments to Author:

Reviewer: 2

Comments to the Author(s)

I feel that the authors have now provided a comprehensive and convincing response to the remaining issue that I had raised in my previous review, so I do not believe that any further changes or additions are required

Reviewer: 3

Comments to the Author(s)

The main dispute related to the use of corrected versions of asymptotic confidence interval based on normality and those from profile likelihood methods is addressed in the revised version.

The uncertainty surrounding estimating the maximum value of a given process still appears to me to be tremendous, but if necessarily needed to be estimated, then I find the methods presented in the current version useful. Alternative and useful ways might also be to report or support decisions based on return levels with finite return periods.

The new edits and changes are accurate and I am happy to recommend the paper to the journal.

Appendix A

Review of “Operating at the extreme: estimating the upper yield boundary of winter wheat production in commercial practice”, manuscript ID RSOS-190242

The authors of this paper successfully use extreme-value theory to establish an upper bound for the yield of winter wheat. The paper is very clear and the analyses are sound. Some suggestions and comments that might deepen the statistical methodology further are found below.

page 2, lines 4–14 These paragraphs might suggest that your analysis can be used to forecast the upper bound of wheat yields. However, the estimated endpoints will probably change radically when exterior conditions change (global temperature increase, demand for food). I think it is important to point out from the beginning that your results are only valid in a stationary world.

page 6, lines 5–10 Please mention that the MLE is only valid for $\gamma > -0.5$.

page 6, lines 31 Maybe write t_k , since t depends on k just like the GPD parameters?

pages 8–10 You chose to stratify your data into geographical regions or agricultural input and to re-run the same analysis. Wouldn't it be more natural to fit all data to a GPD whose parameters depend on covariates? For instance, you could set

$$\gamma(s) = \begin{cases} \gamma_1 & \text{if } s \in \text{East England} \\ \gamma_2 & \text{if } s \in \text{North England} \\ \gamma_3 & \text{if } s \in \text{West England or Wales} \end{cases}$$

while keeping the scale parameter constant (or inversely). Even more interesting, you could let σ or γ vary smoothly as a function of agriculture input (see, for instance, Chavez-Demoulin, Embrechts, and Hofert. (2016). *An extreme value approach for modeling operational risk losses depending on covariates*. Journal of Risk and Insurance 83(3)).

This type of approach would allow you to choose the best model using AIC and give a reliable answer to the question whether γ changes with regional and/or agricultural variations. Moreover, your standard errors will probably be lower as you use the entire sample.

page 13, Table 1 Could you please provide confidence intervals for σ ? You could also provide the asymptotic variance of $\hat{\sigma}_k$ at the top of page 6, together with the asymptotic covariance of $(\hat{\sigma}_k, \hat{\gamma}_k)$.

Appendix B

Referee's report: Operating at the extreme: estimating the upper yield boundary of winter wheat production in commercial practice

Overview and general comments

The paper examines and estimates maximum crop yield in UK using univariate extreme value statistics and specifically, peaks over threshold analysis. The analysis is performed on data collected by the Farm Business Survey for the period 2006-2015. The motivation of the analysis is based on a series of articles that provide evidence of stagnation in productivity and the authors attempt to quantify this by exploiting statistical properties of the tail decay of yield (measured by spatial averages: tonnes per hectare). The analysis suggests finite upper end points on yield distributions and presents a range of tail estimates for spatial stratifications and spending on input levels measured by fertilisers.

The paper is short and easy to read but I have strong concerns on the statistical findings and in particular, on the use of asymptotic approximations for confidence intervals. Attention is also required to the presentation of results and the reporting of tail indices. The results presented are interesting but in its current state the paper does not seem to be sufficiently developed to be a good fit for the journal. The following comments motivate my decision.

Major

- **Return levels vs tail indices:** Reporting estimates of location, scale or tail indices obtained from an extreme value analysis is cumbersome. Instead, it is better practice to include return levels and their associated uncertainty as these have a direct interpretation and facilitate readability by non-experts.
- **Spatial averaging:** I wonder what is the added benefit from an extreme value analysis in this application. As far as I can understand, the data are spatial averages and hence it is not surprising to see negative tail indices. This indicates that a normal distribution for the averages would be reasonable approximation for the entire distribution of yield and not just the extreme part. It would be useful to have a comparison between return levels estimates obtained from threshold exceedances and from a Gaussian model for the entire distribution.
- **Confidence intervals:** I am surprised by the use of equation (4) for approximate 95% confidence intervals. I can appreciate the elegance of the asymptotic approximation yet I am sceptical about its use especially when the data are short-tailed and/or when they are analyzed in various stratifications. The paper would strongly benefit by giving clear guidelines of when such intervals can be used and when not. Comparing normal based intervals with standard profile likelihood based intervals (see Coles (2001)) would be a valuable addition to the paper.

Minor

- pg 4. It would be helpful to include more information in the caption of Figure 1 such as the sample size and the geographical area under study.
- pg 6. "In particular, under standard conditions". Explain what these conditions are about.
- pg 6. What is the approximate distribution of $\hat{\sigma}$?
- pg 6. When a result is sourced from a book, it is helpful to include the number of the page containing that very result. For example when referencing [42] in Equation (6).

- pg 6. Here k is used to denote the k highest yields but later is called *the effective sample size*. It would be helpful to term k the effective sample size the first time you introduce it.
- pg 7. typo “along of”.
- pg 14. Capitalize names only. For example: Generalised Pareto, Probability Weighted Moment, Moment estimator should read generalised Pareto, probability weighted Moment, moment estimator.
- pg 15. typo “on top winter wheat yields”
- reference to `evir` package is not included in section References.

References

Coles, S. G. (2001), *An Introduction to Statistical Modeling of Extreme Values*, Springer–Verlag, London.

Appendix C

TO:

Professor Ruth King and Professor Mark Chaplain
Associate Editor and Subject Editor for *Royal Society Open Science*

RE: Decision regarding submission RSOS-190242 entitled “Operating at the extreme: estimating the upper yield boundary of winter wheat production in commercial practice”

Dear Professor King and Professor Chaplain,

Thank you for your decision email of 23rd May 2019 concerning the above paper. We thank you for the opportunity to revise our manuscript for its publication in Royal Society Open Science. We would like to thank the referees for the careful reading and their valuable remarks and suggestions. The revised version of the manuscript addresses all of these constructive comments. We have added acknowledgements for the referees’ comments in the new version of the paper before the references.

Before our point-by-point response to these comments, we would like to list the main changes we have made for this new version:

- It is now clearly emphasised that our statistical analysis is carried out under an assumption of stationarity, i.e. under current climate and growing conditions.
- We now include confidence intervals for the scale parameter. We have also reworked our construction of the confidence intervals for the shape parameter to account for the fact that the lower bound of the Gaussian confidence interval is unreasonably conservative.
- We have amended our Discussion and conclusions section, by mentioning the possibility of modelling high yields using more modern methodologies linking the distribution parameters to input level or geographical coordinates using smooth functions.

Modified parts in the revised version of the manuscript are marked in red. Below, we have listed a more detailed account on the changes and various issues raised in the reports. We give point-by-point replies to all the comments made by the reviewers. We hope that you will be satisfied with the present version.

Please feel free to contact us if you require any further information. We look forward to hearing from you soon.

Best regards,

Emily G. Mitchell
University of Nottingham

Reply to the AE's comments

Thank you very much for your suggestions and comments. The revised version of the manuscript takes all of them into account. For your convenience we incorporate your comments in italics followed by our reply. Modified parts in the revised version of the manuscript are marked in red.

Comments

The manuscript has been reviewed by three referees and myself. The paper overall is very well written with a clear motivation and application to real data. However, a number of significant issues have been identified within the analysis that raise particular concerns. The primary issues raised relate to:

– modelling considerations - as is the case with any applied problem there will be multiple ways the problem can be approached - and everyone will have a slightly different ideas. However - in this case the referees do provide some useful suggestions regarding the model which I think could significantly improve some of the aspects of the paper - particularly in relation to the spatial aspect and assessment of differences between the regions in a more formal manner particularly given that this is one of the focuses of the paper.

Answer: We have tried to fit the constant scale model suggested by Referee 1, but found that this resulted in poor model fits (see our answer to the fourth comment of Referee 1). The approach where the shape and scale parameters vary smoothly as a function of input or geographical coordinates is very interesting but arguably far out of the scope of our manuscript, whose idea is rather to give the first insight of its kind into the extreme value analysis of winter wheat yield levels. This potentially more powerful approach is now mentioned in the Discussion and conclusions section of the revised version of our manuscript as part of our future research.

Likelihood ratio tests have been implemented to assess whether the appropriate model for the description of high yields depends on geographical region or input use. We find that this is indeed the case, and explain why in our reply to the second minor comment of Referee 2. We do mention these tests in our revised version and what they allow us to conclude, but we do not explicitly report the numerical results for the sake of brevity.

Finally, following the second major comment of Referee 3, we investigated whether a Gaussian model would be appropriate to model our yield data. We find, using visual statistical checks, that the data is generally not adequately described by the Gaussian distribution, especially in the tails. Our interpretation is that the data is made of $n = 1,536$ recordings for individual farms and as such cannot be considered to be spatial averages, so should not be a priori considered to be Gaussian. More generally, there does not appear to be an obvious satisfactory distributional family that would allow for the modelling of yield across regions and levels of input use. This is where there is a crucial benefit from using an extreme value analysis in this application: even though there is no clear parametric model that would describe the distribution of yield, extreme value theory provides a flexible and valid class of models for the high yields that are the focus of the manuscript.

– the exact extracted data that is fitted within the analysis could have additional variability due to the length of time the farms are in the study. This should be considered further and the robustness of this investigated.

Answer: We acknowledge that we had not commented on the statistical consequences of how the data was constructed upon the estimation of the maximum value of yield. Our reply

to the related major comment of Referee 2 explains why, under reasonable assumptions, the distributional model appropriate to the description of the extremes of yield over a single year is also the model appropriate to the description of the extremes of yield over multiple years. As far as statistical inference is concerned, our construction of the data therefore does not create any issue. In fact, in terms of bias, using multiple-year data is intuitively better than single-year data, since their values will tend to be larger and therefore closer to the true value of the endpoint.

– the (approximate) confidence intervals that are used lead to contradictory intervals (in terms of lower limits being below the observed data) - these should be investigated further and the referees provide a useful suggestion here with regard to alternative approaches, particularly in relation to a profile likelihood approach.

Answer: On these data, the lower limits of the 95% confidence intervals for the upper endpoint are lower than the maximum observed value in the data used to derive these estimates. This happens because the Gaussian confidence interval has a lower bound that is not constrained to be larger than the maximum value in the sample (which is a clear lower bound for the true value of the endpoint). As we point out in our replies to Referees 2 and 3, we investigated the use of the profile likelihood method as an alternative approach to the Gaussian construction, but we argue that the results it produces are not satisfactory in our context for physical and biological reasons. With this in mind, and to account for the fact that the Gaussian confidence interval may produce an unreasonably conservative lower bound, we decided to use in the revised version a modified confidence interval. This new construction, and the justification behind it, is given on pages 6–7 of the revised version.

Reply to the comments of Referee 1

Thank you very much for your suggestions and comments. The revised version of the manuscript takes all of them into account. For your convenience we incorporate your comments in italics followed by our reply. Modified parts in the revised version of the manuscript are marked in red.

Overview

The authors of this paper successfully use extreme-value theory to establish an upper bound for the yield of winter wheat. The paper is very clear and the analyses are sound. Some suggestions and comments that might deepen the statistical methodology further are found below.

Comments

page 2, lines 4–14 *These paragraphs might suggest that your analysis can be used to forecast the upper bound of wheat yields. However, the estimated endpoints will probably change radically when exterior conditions change (global temperature increase, demand for food). I think it is important to point out from the beginning that your results are only valid in a stationary world.*

Answer: You are correct. It is now clearly pointed out in the Abstract that our extreme value analysis is carried out under current climate and growing conditions. This emphasis on current growing conditions is also repeated at the end of the second paragraph of the Introduction.

page 6, lines 5–10 *Please mention that the MLE is only valid for $\gamma > -0.5$.*

Answer: This is now mentioned just before Equation (2) stating the asymptotic normality of the MLE.

page 6, lines 31 *Maybe write t_k , since t depends on k just like the GPD parameters?*

Answer: Done.

pages 8–10 *You chose to stratify your data into geographical regions or agricultural input and to re-run the same analysis. Wouldn't it be more natural to fit all data to a GPD whose parameters depend on covariates? For instance, you could set*

$$\gamma(s) = \begin{cases} \gamma_1 & \text{if } s \in \text{East England} \\ \gamma_2 & \text{if } s \in \text{North England} \\ \gamma_3 & \text{if } s \in \text{West England or Wales} \end{cases}$$

*while keeping the scale parameter constant (or inversely). Even more interesting, you could let σ or γ vary smoothly as a function of agriculture input (see, for instance, Chavez-Demoulin, Embrechts, and Hofert. (2016). An extreme value approach for modeling operational risk losses depending on covariates. *Journal of Risk and Insurance* 83(3)).*

This type of approach would allow you to choose the best model using AIC and give a reliable answer to the question whether γ changes with regional and/or agricultural variations. Moreover, your standard errors will probably be lower as you use the entire sample.

Answer: Our rationale for fitting individual generalised Pareto models to the different groups of data (stratified geographically or by input level) was to obtain accurate parametric models

without making strong assumptions about the kind of dependence existing between yield and the stratifying variable.

Following your advice, we have tried fitting a constant scale model with varying shape parameter to the data stratified geographically. This was done using the `fevd` command, part of the `extRemes` R package, and yields a global scale estimate $\hat{\sigma} = 1.37$, with stratified scale estimates of $\hat{\gamma}_{\text{North}} = -0.31$, $\hat{\gamma}_{\text{West}} = -0.16$ and $\hat{\gamma}_{\text{East}} = -0.06$.

To assess the quality of model fit, we represented in Figure 1 QQ-plots of the data versus this stratified model, along with QQ-plots for the individual generalised Pareto models in our three identified regions. It can be seen in this Figure that while the individual generalised Pareto models overall describe well the stratified data, the constant scale assumption results in a poor model fit. This makes us prefer the individual generalised Pareto models we had fit in the first version of the manuscript.

The idea of letting the scale or shape parameter (or both) vary smoothly as a function of input level or geographical coordinates, as in Chavez-Demoulin *et al.* (2016), is very interesting. It is, however, arguably far out of the scope of our manuscript, whose idea is rather to give the first insight of its kind into the extreme value analysis of winter wheat yield levels. A more refined subsequent analysis, whose focus would be the accurate prediction of maximal yield levels as a function of agricultural input use and location, would undoubtedly benefit from this kind of precise conditional analysis. This is now mentioned in the Discussion and conclusions section of the revised version of our manuscript.

page 13, Table 1 *Could you please provide confidence intervals for σ ? You could also provide the asymptotic variance of $\hat{\sigma}_k$ at the top of page 6, together with the asymptotic covariance of $(\hat{\sigma}_k, \hat{\gamma}_k)$.*

Answer: Confidence intervals for σ are now provided in Table 1. They are constructed using the joint asymptotic normality of $(\hat{\gamma}_k, \hat{\sigma}_k)$, which is now stated in Equation (2).

Figure 1: QQ-plots of model fits for the data stratified geographically. From left to right: constant scale generalised Pareto model applied to the whole data set; generalised Pareto model for North England and Wales; generalised Pareto model for East England.

Reply to the comments of Referee 2

Thank you very much for your suggestions and comments. The revised version of the manuscript takes all of them into account. For your convenience we incorporate your comments in italics followed by our reply. Modified parts in the revised version of the manuscript are marked in red.

Overview

Overall, I found this to be a clear, concise and very well written paper, that provides an novel, interesting and practically relevant application of a relatively sophisticated form of statistical modelling. The paper has a clear application, the exposition of the methodology is good, and the level of technical detail seemed to me to be appropriate. I thought that the use of Extreme Value Theory to address this problem was defensible, worthwhile and interesting. The discussion of the outputs, in relation to agricultural production, seemed very clear, and I thought that this constituted an accurate reflection of the findings of the analysis. The text also, I felt, properly acknowledged the caveats and limitations associated with the approach used.

I feel, however, that one major revision, and a few minor revisions, are needed in order to ensure that the selection of methods and the interpretation of the results are fully defensible.

Proposed major revision

A. In situations where a farm has data over multiple years, the authors have selected the maximum yield for that farm, across all years, to go into the analysis. Does this approach not create potential for bias, since the maximum yield per farm will tend to be positively related to the number of years used in constructing the maximum value? (i.e., all else being equal, the maximum observed yield for a farm with 5 years of data will tend on average to be systematically larger than the maximum yield for a farm with two years of data...). If that is the case, I think some evaluation of the likely consequences of this is needed (e.g. by adapting the model to account for differences in the number of years, and/or by assessing whether the results change qualitatively when an alternative approach to data selection which avoids this issue - such as randomly selecting a single year of data for each farm - is used).

Answer: It is indeed the case that we did not comment on the statistical consequences of how the data was constructed upon the estimation of the maximum value of yield. The purpose of our argument below is to show that, under reasonable assumptions, the distributional model appropriate to the description of the extremes of yield over a single year is also the model appropriate to the description of the extremes of yield over multiple years. As far as statistical inference is concerned, our construction of the data therefore does not create any issue.

Let Y be a random variable having the distribution of the one-year yield for a farm (over the whole of England and Wales). For a given farm in our sample, yields are recorded over N years, where $N \in \{1, \dots, 10\}$ is a discrete random variable, and we instead take the maximum such yield as an individual data point. In other words, our data is made of $n = 1,536$ data points whose distribution is that of $X = \max_{1 \leq i \leq N} Y_i$, where the Y_i are independent copies of Y . We suppose in what follows that N is independent of Y and thus of X . It should, first, be clear that X and Y have the same support and in particular the same right endpoint x^* . We then have, for any $t < x^*$,

$$\forall y > 0, \mathbb{P}(X - t \leq y \mid X > t) = \frac{\mathbb{P}(\max_{1 \leq i \leq N} Y_i \leq y + t) - \mathbb{P}(\max_{1 \leq i \leq N} Y_i \leq t)}{1 - \mathbb{P}(\max_{1 \leq i \leq N} Y_i \leq t)}.$$

By independence of N and the Y_i , the numerator above can be rewritten

$$\begin{aligned}
& \mathbb{P}\left(\max_{1 \leq i \leq N} Y_i \leq y+t\right) - \mathbb{P}\left(\max_{1 \leq i \leq N} Y_i \leq t\right) \\
&= \sum_{m=1}^{10} \left[\mathbb{P}\left(\max_{1 \leq i \leq m} Y_i \leq y+t\right) - \mathbb{P}\left(\max_{1 \leq i \leq m} Y_i \leq t\right) \right] \mathbb{P}(N=m) \\
&= \sum_{m=1}^{10} [\{\mathbb{P}(Y \leq y+t)\}^m - \{\mathbb{P}(Y \leq t)\}^m] \mathbb{P}(N=m) \\
&= [\mathbb{P}(Y \leq y+t) - \mathbb{P}(Y \leq t)] \sum_{m=1}^{10} \mathbb{P}(N=m) \sum_{j=0}^{m-1} \{\mathbb{P}(Y \leq y+t)\}^j \{\mathbb{P}(Y \leq t)\}^{m-1-j}.
\end{aligned}$$

When t is high, and regardless of the value of $y > 0$, $\mathbb{P}(Y \leq y+t)$ and $\mathbb{P}(Y \leq t)$ are close to 1; in other words, we have

$$\begin{aligned}
\mathbb{P}\left(\max_{1 \leq i \leq N} Y_i \leq y+t\right) - \mathbb{P}\left(\max_{1 \leq i \leq N} Y_i \leq t\right) &\approx [\mathbb{P}(Y \leq y+t) - \mathbb{P}(Y \leq t)] \sum_{m=1}^{10} m \mathbb{P}(N=m) \\
&= [\mathbb{P}(Y \leq y+t) - \mathbb{P}(Y \leq t)] \mathbb{E}(N)
\end{aligned}$$

when t is high, for any $y > 0$. Similarly

$$1 - \mathbb{P}\left(\max_{1 \leq i \leq N} Y_i \leq t\right) \approx [1 - \mathbb{P}(Y \leq t)] \mathbb{E}(N)$$

when t is high, and therefore

$$\begin{aligned}
\forall y > 0, \mathbb{P}(X - t \leq y | X > t) &\approx \frac{\mathbb{P}(Y \leq y+t) - \mathbb{P}(Y \leq t)}{1 - \mathbb{P}(Y \leq t)} \\
&= \mathbb{P}(Y - t \leq y | Y > t).
\end{aligned}$$

This implies that the generalised Pareto model appropriate to model the right tail of the one-year yield Y is also a sensible model for the right tail of our multiple-year maximum X . In terms of bias, using multiple-year data is actually intuitively better than single-year data, since their values will tend to be larger and therefore closer to the true value of the right endpoint.

Proposed minor revisions

1. I was rather surprised that in many cases (if I am reading it correctly) the lower limits of the 95% confidence intervals of the estimate for the upper endpoint were lower than the maximum observed value in the data used to derive these estimates, and in some cases seemed much lower. This seems counter-intuitive, given that the upper endpoint must presumably, in reality, be no lower than the maximum observed value. Why does this occur, and does the fact that it occurs indicate an issue with the model/inference? I think the authors need to comment on this, and, if they are convinced that this is not an issue, provide a justification for that.

(More specifically, does this phenomenon arise from the normality assumption in Equations 2 and 4, and if so could it indicate an issue with that assumption? [e.g. the estimator is presumably only asymptotically normal, so could that assumption be breaking down substantially for samples of this size?]. Is there any possibility to avoid this issue by using an alternative approach to construction of the confidence interval - e.g. profile likelihood, as I think is used

in Coles (2001) and the associated “ismev” R package, which I believe avoids an assumption of symmetry.)

Answer: You are correct in writing that, on these data, the lower limits of the 95% confidence intervals for the upper endpoint are lower than the maximum observed value in the data used to derive these estimates. This happens because the Gaussian confidence interval has a lower bound that is not constrained to be larger than the maximum value in the sample (a clear lower bound for the true value of the endpoint). This is especially important when the estimate $\hat{\gamma}_k$ is close to zero, which is a regular occurrence in our analyses, because then the presence of the factor $1/\hat{\gamma}_k^2$ in Equation (4) of the manuscript typically makes this lower bound unreasonably conservative.

We investigated the use of profile likelihood as an alternative approach to the Gaussian construction. This was done using the `gpd.prof` and `gpd.profxi` commands part of the R package `ismev`, as you suggested. The 95% profile likelihood confidence interval for the shape parameter is represented in Figure 2. It can be seen in this Figure that the confidence interval overlaps 0. In the framework of profile likelihood, this implies that, as the level of the return increases, profile likelihoods for return levels get flatter and therefore that the upper bound of the confidence interval tends to infinity, as is illustrated on Figure 3, indicating the uncertainty about the type of tail (bounded or unbounded) the data exhibits. The bottom right panel of Figure 3 represents the profile likelihood calculated by `gpd.prof` for right endpoint estimation: clearly, the method fails here to produce a confidence interval that is bounded to the right, and it is arguable whether it gives a satisfactory numerical result at all.

We do not think that this conclusion provided by the profile likelihood method is sensible in our context of statistical analysis of wheat yield. Indeed, wheat production in the UK’s temperate climate is characterised by high input-high output biological relationships, with farmers applying high input levels of nitrogen as they aim to produce for high yield (see Reader *et al.*, 2018) rather than economically optimal yields. It is also known that biological cropping systems, such as winter wheat farms, typically exhibit diminishing productivity functions with respect to input-output relationships (see Tilman *et al.*, 2002 and Mueller *et al.*, 2014), and even more strongly, that an over-application of inputs can lead to marginal yield reductions. In the context of the agricultural input-intensive UK commercial production of wheat, this means that the high levels of wheat yields that we observe in our data are quite likely to be of the order of magnitude of maximum wheat yield; in any event, it is very unlikely that an infinitely large yield is physically and biologically possible (a related point is made by Yoshida, 1972).

With this in mind, and to account for the fact that the Gaussian confidence interval may produce an unreasonably conservative lower bound, we decided to use in the revised version the confidence interval

$$\left[\max \left(t_0, \hat{x}_k^* - \frac{1.96}{\sqrt{k}} \times \frac{\hat{\sigma}_k}{\hat{\gamma}_k^2} \times \sqrt{1 + 4\hat{\gamma}_k + 5\hat{\gamma}_k^2 + 2\hat{\gamma}_k^3 + 2\hat{\gamma}_k^4} \right), \right. \\ \left. \hat{x}_k^* + \frac{1.96}{\sqrt{k}} \times \frac{\hat{\sigma}_k}{\hat{\gamma}_k^2} \times \sqrt{1 + 4\hat{\gamma}_k + 5\hat{\gamma}_k^2 + 2\hat{\gamma}_k^3 + 2\hat{\gamma}_k^4} \right]$$

as an approximate 95% confidence interval for the maximum yield x^* , where t_0 denotes the maximum value in the sample. The upper bound of this interval is exactly the upper bound produced by the Gaussian approximation in Equation (4). The lower bound, meanwhile, is equal to the maximum observation in the sample or the lower bound produced by the Gaussian approximation, whichever is greater. [Note that truncating the interval at level t_0 does not affect its coverage probability in practice because, by definition, the true value x^* of the right endpoint must be larger than t_0 with probability 1.] This interval allows us to give a sensible

lower bound for the right endpoint of wheat yield, and at the same time provide a realistic upper bound for the confidence interval. This new construction, and the justification behind it, is given on pages 6–7 of the revised version.

2. *Would it be possible to formally test whether the differences between regions, and levels of input, are significant? - e.g. using a likelihood ratio test? It seems slightly unsatisfactory that at the moment the only way to evaluate whether the differences are significant are not is to look at whether the confidence intervals for groups overlap each other (which is not, I think, equivalent to a formal test of a significant difference between regions/input levels).*

Answer: You are correct in pointing out that searching for overlapping confidence intervals for model parameters is not equivalent to formally testing whether distributions are different. We followed your advice and used a likelihood ratio test to evaluate the discrepancy in our models between regions and levels of input. For two different regions/levels of input A and B, with associated high levels of yield \mathbf{x}_A and \mathbf{x}_B , we used the `fevd` command from the `extRemes` R package to:

- (i) Fit two generalised Pareto models separately to the two groups A and B using maximum likelihood. This results in two estimates $(\hat{\gamma}_A, \hat{\sigma}_A)$ and $(\hat{\gamma}_B, \hat{\sigma}_B)$. We then calculate separately the two maximum likelihoods $L(\hat{\gamma}_A, \hat{\sigma}_A|\mathbf{x}_A)$ and $L(\hat{\gamma}_B, \hat{\sigma}_B|\mathbf{x}_B)$. We finally compute $L = L(\hat{\gamma}_A, \hat{\sigma}_A|\mathbf{x}_A) \times L(\hat{\gamma}_B, \hat{\sigma}_B|\mathbf{x}_B)$. This represents the maximum likelihood under the full model, with 4 parameters, describing A and B jointly.
- (ii) Fit a single generalised Pareto model to the combined data $\mathbf{x}_{AUB} = (\mathbf{x}_A, \mathbf{x}_B)$ using maximum likelihood. This results in an estimate $(\hat{\gamma}_{AUB}, \hat{\sigma}_{AUB})$. We then compute the corresponding maximum likelihood $L_0 = L(\hat{\gamma}_{AUB}, \hat{\sigma}_{AUB}|\mathbf{x}_{AUB})$. This represents the maximum likelihood under the restricted model where A and B can be described by the same distribution.

The relevant likelihood ratio test statistic for testing the null hypothesis of equal models $H_0 : (\gamma_A, \sigma_A) = (\gamma_B, \sigma_B)$ is then the deviance $D = -2 \log(L_0/L)$, to be compared to the 95% quantile of the χ^2 distribution with $4 - 2 = 2$ degrees of freedom, equal to 5.99. Results are reported in Table 1.

Case	Deviance D	Conclusion
A = North England, B = West England & Wales	107	Reject H_0
A = North England, B = East England	262	Reject H_0
A = West England & Wales, B = East England	240	Reject H_0
A = Low input, B = Medium input	113	Reject H_0
A = Low input, B = High input	173	Reject H_0
A = Medium input, B = High input	71.7	Reject H_0

Table 1: Likelihood ratio test statistics

These results therefore confirm that the extreme value models to be used for the description of high yield levels differ depending on region and level of agricultural input use. This is now briefly mentioned on page 9 of the revised version, although we do not report either the construction of the test or the above table for the sake of brevity.

3. *The statement “maximum yield levels appear to plateau as a function of crop protection and fertiliser use” is made in the abstract, but I couldn’t follow the rationale for this conclusion - the difference between the point estimate of the upper endpoint of yield under the “high*

inputs” category and “medium inputs category” seems to be larger than the difference between the endpoints under the “medium input” and “low input” categories, so I couldn’t see that the results of the analysis provided evidence for a plateau? (I certainly agree that the results could be consistent with the existence of a plateau, but I don’t quite follow the rationale for how they provide positive evidence for the existence of one, especially since the uncertainty associated with the estimated upper endpoint for each input level is very large).

Answer: This was indeed an overstatement. It is now replaced by the statement “neither is there statistical evidence that maximum yield levels are improved by high levels of crop protection and fertiliser use.” This is consistent with our revision following your point 4 below.

4. “does not suggest that spending a larger amount of capital on fertiliser and crop protection significantly improves”: I think the wording here rather confuses statistical significance and agricultural significance. I would suggest that rephrasing this to “does not provide significant evidence that spending a larger amount of capital on fertiliser and crop protection improves” provides a more accurate wording.

Answer: Done, thanks.

5. Is Equation 4 conditional on the shape parameter being negative? If so, will it tend to underestimate uncertainty in situations where the confidence interval for the shape parameter overlaps zero? If that is the case, I think that this is not necessarily a problem (especially since in this application the confidence intervals rarely do overlap zero) but that it should be explicitly noted in the text as a caveat associated with these confidence intervals.

Answer: Equation (4) is indeed conditional on the shape parameter being negative, and this is now clearly emphasised just before this equation. In the case when the confidence interval for the shape parameter contains 0, the calculation of the confidence interval will indeed return lower uncertainty, compared to the profile likelihood interval, in computing the upper bound of the confidence interval. This is now mentioned in the revised version below Equation (4). As we write in our answer above to your comment 1, it is, however, arguable that an infinite upper bound for the maximum yield confidence interval, which is what would be returned for these data by the profile likelihood method, is not sensible in our context of maximum yield estimation.

Figure 2: Full yield data set, graph of the profile likelihood function for the shape parameter γ . The two intersections of the concave full line with the second-from-top horizontal full line define the 95% profile likelihood confidence interval for γ . The vertical dashed line is the line $\gamma = 0$.

Figure 3: Full yield data set of size $n = 1536$, graphs of the profile likelihood functions for return levels. Top left: return level with exceedance probability $1/n$, top right: return level with exceedance probability $1/(10n)$, bottom left: return level with exceedance probability $1/(100n)$, bottom right: maximum yield (informally, return level with exceedance probability $1/\infty = 0$).

Reply to the comments of Referee 3

Thank you very much for your suggestions and comments. The revised version of the manuscript takes all of them into account. For your convenience we incorporate your comments in italics followed by our reply. Modified parts in the revised version of the manuscript are marked in red.

Overview and general comments

The paper examines and estimates maximum crop yield in UK using univariate extreme value statistics and specifically, peaks over threshold analysis. The analysis is performed on data collected by the Farm Business Survey for the period 2006-2015. The motivation of the analysis is based on a series of articles that provide evidence of stagnation in productivity and the authors attempt to quantify this by exploiting statistical properties of the tail decay of yield (measured by spatial averages: tonnes per hectare). The analysis suggests finite upper end points on yield distributions and presents a range of tail estimates for spatial stratifications and spending on input levels measured by fertilisers.

The paper is short and easy to read but I have strong concerns on the statistical findings and in particular, on the use of asymptotic approximations for confidence intervals. Attention is also required to the presentation of results and the reporting of tail indices. The results presented are interesting but in its current state the paper does not seem to be sufficiently developed to be a good fit for the journal. The following comments motivate my decision.

Major

Return levels vs tail indices: Reporting estimates of location, scale or tail indices obtained from an extreme value analysis is cumbersome. Instead, it is better practice to include return levels and their associated uncertainty as these have a direct interpretation and facilitate readability by non-experts.

Answer: We agree with you that return levels are far easier to explain and interpret than estimated model parameters. A special case of return level is the right endpoint; as a matter of fact, the right endpoint x^* of a distribution function F is defined as $x^* = \inf\{t \in \mathbb{R} \mid F(t) \geq 1\}$, and can as such be considered to be the 100% return level associated to the distribution function F . For a random variable X with distribution function F , the right endpoint x^* can equivalently be interpreted as the maximum possible value of X . We would like to highlight that our main objective in the present paper is precisely to estimate maximum possible values of winter wheat yield under current growing conditions in England and Wales. This is underlined in the title of our paper (“estimating the upper yield boundary of winter wheat production”), in our Introduction section (see, for instance, the end of the fourth paragraph therein) and throughout the Results section, where most of the discussion centres around the estimated values of maximum yield and whether they can be thought to differ across regions and level of input use. Our Discussion and conclusions section provides further perspectives on, and interpretation of, our maximum yield estimates. It is indeed the case that we briefly report parameter estimates early in the Results section and in Table 1; this should be seen as a way to help the reader unfamiliar with extreme value analysis understand that our maximum yield estimates are reached by combining the threshold, scale and shape parameter estimates. With this in mind, and to clarify further the scope of the paper, we have:

- Added at the end of the Introduction that our account of the results is given “with an emphasis on estimated maximum yield levels”;
- Emphasised at the beginning of the Results section that our objective was to “estimate the maximum value of yield”;
- Clearly indicated the dependence of maximum yield estimates upon the threshold, scale and shape parameter estimates in Table 1. We also rewrote its caption to make clear that maximum yield level estimates are the main information within the table.

Spatial averaging: I wonder what is the added benefit from an extreme value analysis in this application. As far as I can understand, the data are spatial averages and hence it is not surprising to see negative tail indices. This indicates that a normal distribution for the averages would be reasonable approximation for the entire distribution of yield and not just the extreme part. It would be useful to have a comparison between return levels estimates obtained from threshold exceedances and from a Gaussian model for the entire distribution.

Answer: As we mention in our Data and methods section (see pp.3-4), the data is made of $n = 1,536$ recordings for individual farms. As such, the data cannot be considered to be spatial averages (because it is not made of values averaged over large regions) and therefore should not be a priori considered to be Gaussian. To check whether the Gaussian model was appropriate we, for each of our studied cases (full sample of data, data stratified by region and data stratified by agricultural input use), ran visual checks composed of:

- A histogram of the data, superimposed with the Gaussian fit obtained via maximum likelihood,
- A kernel density fit produced using the standard `density` R command, superimposed with this same Gaussian fit,
- A Gaussian QQ-plot of the centred and standardised data.

These plots are given in Figures 4–10 below. It can be seen from this series of plots that the data is generally not adequately described by the Gaussian distribution, especially in the tails. More generally, the kernel density fits do not appear to suggest an obvious satisfactory distributional family that would allow for the modelling of yield across regions and levels of input use. In our opinion, this is where there is a crucial benefit from using an extreme value analysis in this application: even though there is no clear parametric model that would describe the distribution of yield, extreme value theory provides a flexible and valid class of models for the high yields that are the focus of the manuscript.

Confidence intervals: I am surprised by the use of equation (4) for approximate 95% confidence intervals. I can appreciate the elegance of the asymptotic approximation yet I am sceptical about its use especially when the data are short-tailed and/or when they are analyzed in various stratifications. The paper would strongly benefit by giving clear guidelines of when such intervals can be used and when not. Comparing normal based intervals with standard profile likelihood based intervals (see Coles (2001)) would be a valuable addition to the paper.

Answer: We investigated the use of profile likelihood as an alternative approach to the Gaussian construction. This was done using the `gpd.prof` and `gpd.profxi` commands part of the R package `ismev`. The 95% profile likelihood confidence interval for the shape parameter is represented in Figure 11. It can be seen in this Figure that the confidence interval overlaps 0. In the framework of profile likelihood, this implies that, as the level of the return increases, profile

likelihoods for return levels get flatter and therefore that the upper bound of the confidence interval tends to infinity, as is illustrated on Figure 12, indicating the uncertainty about the type of tail (bounded or unbounded) the data exhibits. The bottom right panel of Figure 12 represents the profile likelihood calculated by `gpd.prof` for right endpoint estimation: clearly, the method fails here to produce a confidence interval that is bounded to the right, and it is arguable whether it gives a satisfactory numerical result at all.

We do not think that this conclusion provided by the profile likelihood method is sensible in our context of statistical analysis of wheat yield. Indeed, wheat production in the UK's temperate climate is characterised by high input-high output biological relationships, with farmers applying high input levels of nitrogen as they aim to produce for high yield (see Reader *et al.*, 2018) rather than economically optimal yields. It is also known that biological cropping systems, such as winter wheat farms, typically exhibit diminishing productivity functions with respect to input-output relationships (see Tilman *et al.*, 2002 and Mueller *et al.*, 2014), and even more strongly, that an over-application of inputs can lead to marginal yield reductions. In the context of the agricultural input-intensive UK commercial production of wheat, this means that the high levels of wheat yields that we observe in our data are quite likely to be of the order of magnitude of maximum wheat yield; in any event, it is very unlikely that an infinitely large yield is physically and biologically possible (a related point is made by Yoshida, 1972).

With this in mind, it is also correct to point out that the Gaussian approximation has its shortcomings. For instance, on these data, the lower limits of the 95% confidence intervals for the upper endpoint are lower than the maximum observed value in the data used to derive these estimates. This happens because the Gaussian confidence interval has a lower bound that is not constrained to be larger than the maximum value in the sample (a clear lower bound for the true value of the endpoint). This is especially important when the estimate $\hat{\gamma}_k$ is close to zero, because then the presence of the factor $1/\hat{\gamma}_k^2$ in Equation (4) of the manuscript typically makes this lower bound unreasonably conservative. To account for that, we decided to use in the revised version the confidence interval

$$\left[\max \left(t_0, \hat{x}_k^* - \frac{1.96}{\sqrt{k}} \times \frac{\hat{\sigma}_k}{\hat{\gamma}_k^2} \times \sqrt{1 + 4\hat{\gamma}_k + 5\hat{\gamma}_k^2 + 2\hat{\gamma}_k^3 + 2\hat{\gamma}_k^4} \right), \right. \\ \left. \hat{x}_k^* + \frac{1.96}{\sqrt{k}} \times \frac{\hat{\sigma}_k}{\hat{\gamma}_k^2} \times \sqrt{1 + 4\hat{\gamma}_k + 5\hat{\gamma}_k^2 + 2\hat{\gamma}_k^3 + 2\hat{\gamma}_k^4} \right]$$

as an approximate 95% confidence interval for the maximum yield x^* , where t_0 denotes the maximum value in the sample. The upper bound of this interval is exactly the upper bound produced by the Gaussian approximation in Equation (4). The lower bound, meanwhile, is equal to the maximum observation in the sample or the lower bound produced by the Gaussian approximation, whichever is greater. This interval allows us to give a sensible lower bound for the right endpoint of wheat yield, and at the same time provide a realistic upper bound for the confidence interval. This new construction, and the justification behind it, is given on pages 6–7 of the revised version.

Minor

pg 4. It would be helpful to include more information in the caption of Figure 1 such as the sample size and the geographical area under study.

Answer: The caption now mentions that the data was collected over the whole of England and Wales, and that the average sample size for each year is 695.

pg 6. *“In particular, under standard conditions”. Explain what these conditions are about.*

Answer: It is now explicitly mentioned just before Equation (2) that asymptotic normality of the MLE holds “when $\gamma > -1/2$ and under a classical second-order condition making it possible to quantify the gap between the actual underlying right tail and the tail of the generalised Pareto distribution.”

pg 6. *What is the approximate distribution of $\hat{\sigma}$?*

Answer: It is Gaussian, and in fact the joint asymptotic distribution of $(\hat{\gamma}_k, \hat{\sigma}_k)$ is Gaussian as well. This is now made clear in Equation (2) of the revised version.

pg 6. *When a result is sourced from a book, it is helpful to include the number of the page containing that very result. For example when referencing [42] in Equation (6).*

Answer: Done.

pg 6. *Here k is used to denote the k highest yields but later is called the effective sample size. It would be helpful to term k the effective sample size the first time you introduce it.*

Answer: Done, see the bottom of page 5.

pg 7. *typo “along of”.*

Answer: Done, now replaced by “along with”.

pg 14. *Capitalize names only. For example: Generalised Pareto, Probability Weighted Moment, Moment estimator should read generalised Pareto, probability weighted Moment, moment estimator.*

Answer: Done.

pg 15. *typo “on top winter wheat yields”*

Answer: Done, now replaced by “upon winter wheat yields”.

reference to evir package is not included in section References.

Answer: We now cite the package as reference [53] B. Pfaff, E. Zivot, A. McNeil, and A. Stephenson. *evir: Extreme Values in R*, 2018. R package version 1.7-4.

Figure 4: Full yield data set, left: histogram and density of Gaussian fit, middle: kernel density fit and Gaussian fit, right: Gaussian QQ-plot of the data.

Figure 5: Yield data set for North England, left: histogram and density of Gaussian fit, middle: kernel density fit and Gaussian fit, right: Gaussian QQ-plot of the data.

Figure 6: Yield data set for West England and Wales, left: histogram and density of Gaussian fit, middle: kernel density fit and Gaussian fit, right: Gaussian QQ-plot of the data.

Figure 7: Yield data set for East England, left: histogram and density of Gaussian fit, middle: kernel density fit and Gaussian fit, right: Gaussian QQ-plot of the data.

Figure 8: Yield data set for low input levels, left: histogram and density of Gaussian fit, middle: kernel density fit and Gaussian fit, right: Gaussian QQ-plot of the data.

Figure 9: Yield data set for medium input levels, left: histogram and density of Gaussian fit, middle: kernel density fit and Gaussian fit, right: Gaussian QQ-plot of the data.

Figure 10: Yield data set for high input levels, left: histogram and density of Gaussian fit, middle: kernel density fit and Gaussian fit, right: Gaussian QQ-plot of the data.

Figure 11: Full yield data set, graph of the profile likelihood function for the shape parameter γ . The two intersections of the concave full line with the second-from-top horizontal full line define the 95% profile likelihood confidence interval for γ . The vertical dashed line is the line $\gamma = 0$.

Figure 12: Full yield data set of size $n = 1536$, graphs of the profile likelihood functions for return levels. Top left: return level with exceedance probability $1/n$, top right: return level with exceedance probability $1/(10n)$, bottom left: return level with exceedance probability $1/(100n)$, bottom right: maximum yield (informally, return level with exceedance probability $1/\infty = 0$).

References

- Chavez-Demoulin, V., Embrechts, P. and Hofert, M. (2016). An extreme value approach for modeling operational risk losses depending on covariates, *Journal of Risk and Insurance* **83**(3): 735–776.
- Coles, S.G. (2001). *An Introduction to Statistical Modeling of Extreme Values*. Springer-Verlag, London.
- Mueller, N.D., West, P.C., Gerber, J.S., MacDonald, G.K., Polasky, S. and Foley, J.A. (2014). A tradeoff frontier for global nitrogen use and cereal production, *Environmental Research Letters* **9**(5): 054002.
- Reader, M.A., Revoredo-Giha, C., Lawrence, R.J., Hodge, I.D. and Lang, B.G. (2018). Farmers' spending on variable inputs tends to maximise crop yields, not profit, *International Journal of Agricultural Management* **7**(1): 1–11.
- Tilman, D., Cassman, K.G., Matson, P.A., Naylor, R. and Polasky, S. (2002). Agricultural sustainability and intensive production practices, *Nature* **418**(6898): 671–677.
- Yoshida, S. (1972). Physiological aspects of grain yield, *Annual Review of Plant Physiology* **23**(1): 437–464.

Appendix D

Referee's report: Operating at the extreme: estimating the upper yield boundary of winter wheat production in commercial practice

Overview and general comments

It is clear that the authors have attempted to carefully revise and respond to the questions posed by all referees. However, I cannot say I am happy with the revision and the methods the authors tried to implement. All referees have tried to help so that the paper implements well founded statistical principles that effectively overcome almost all problems raised in the analysis of the data. The revised version appears to take a strong stance against profile likelihood intervals and suggests a series of modifications to confidence intervals based on asymptotic normality so that problems do not appear anymore (e.g., lower end point of interval being less than observed maximum in the sample, uncertainty in the upper tail not accounted!, etc.). What is even more unsettling is that the paper suggests evidence for boulder upper tail in crop yield but this seems to be "shown" by, effectively, imposing such an assumption, i.e., by using confidence intervals that are bound to be bounded to the right. Given the author's strong prior beliefs on the tail of crop yield, it would seem more natural to follow a Bayesian approach and carefully use prior knowledge rather than implementing ad hoc procedures. I include my comments below.

Major

- The authors are correct in pointing out that profile likelihood based confidence intervals reflect the uncertainty on the type of the upper tail so that when the return period increases, the profile likelihood becomes flatter. One further strength that hasn't been pointed out is that profile likelihood based intervals do not result in erroneous lower endpoints, an issue authors decided to address with an ad hoc procedure that involves an adjustment of the endpoint. What is even more unsettling is the argument against using profile likelihood based intervals: the authors suggest in their response that "it is very unlikely that an infinitely large yield is physically and biologically possible. . . . Clearly the method fails here to produce a confidence interval that is bounded to the right, and it is arguable whether it gives a satisfactory numerical result at all". This argument fails short in several respects as, for example, one can argue in exactly the same way for any physical process. Take rainfall as an example. No matter how much water is evaporated and precipitated, the amount of rainfall will always be finite. However, I have never seen any analysis suggesting intervals bounded to the right be used.
- Observing an infinitely large yield would only be possible had the underlying distribution had an atom at infinity which no extreme value model entertains. So, after reading the author's comment, it is clear that there is a confusion about what $x_i > 0$ means.
- In my previous report I recommended return level plots are presented instead of point and interval estimates of extreme value model parameters. Of course, the maximum of the distribution is a return level but a **return level plot** shows the full tail by plotting return levels against **different return periods**.
- The profile likelihood based intervals in the report to referees suggest there are cases where there isn't sufficient evidence to claim a bounded upper tail as 0 is contained in the interval. Since stratification reduces sample size and increases the uncertainty in the estimates, I would suggest authors think in more detail about how to better model the physical processes so that information is pooled (e.g., across space or otherwise) a point that was raised by another referee. The authors decided to suggest

that this goes beyond the scope of the present paper which in its current form, simply fits extreme value models to strata and chops interval estimates to claim evidence for bounded tails.

Minor

- globally replace “Equation” by “equation”
- pg 6. Replace “Gaussian confidence interval” by “approximate confidence interval based on asymptotic normality”
- pg 6. “This allows approximate confidence intervals for” replace by “This allows approximate confidence intervals based on asymptotic normality for”
- pg 6. “for our maximum yield estimates”. Replace by “for the true maximum yield”. The interval is an estimator of the “true quantity” of interest, not of the point estimate itself.

Appendix E

TO:
Professor Ruth King and Professor Mark Chaplain
Associate Editor and Subject Editor for *Royal Society Open Science*

RE: Decision regarding submission RSOS-191226 entitled “Operating at the extreme: estimating the upper yield boundary of winter wheat production in commercial practice”

Dear Professor King and Professor Chaplain,

Thank you for your decision email of 2nd October 2019 concerning the above paper. We thank you again for the opportunity to revise our manuscript for publication in Royal Society Open Science. We would like to thank the referees for their second careful reading and their valuable remarks and suggestions. The revised version of the manuscript addresses all of these constructive comments.

Modified parts in the revised version of the manuscript are marked in red. We give point-by-point replies to all the comments made by the reviewers. We hope that you will be satisfied with the present version.

Please feel free to contact us if you require any further information. We look forward to hearing from you soon.

Best regards,

Emily G. Mitchell
University of Nottingham

Reply to the AE's comments

Thank you very much for your suggestions and comments. The revised version of the manuscript takes all of them into account. For your convenience we incorporate your comments in italics followed by our reply. Modified parts in the revised version of the manuscript are marked in red.

Comments

The revised manuscript has been reviewed by the same three reviewers as the initial submission. However, there are still significant concerns regarding the revised version of the manuscript, particularly from reviewers 2 and 3, who both feel that their previous comments have not been satisfactorily addressed. For example, this includes, in particular, the construction of the confidence intervals (and associated issues with different approaches), as well as the argument associated with the variations in sample size, and the associated presentation of return levels.

Answer: We have given detailed point-by-point responses to the referees and have taken appropriate actions. We hope that you and the referees will be satisfied with the changes that have been made.

Reply to the comments of Referee 2

Thank you very much for your suggestions and comments. The revised version of the manuscript takes all of them into account. For your convenience we incorporate your comments in italics followed by our reply. Modified parts in the revised version of the manuscript are marked in red.

Overview

The authors have dealt very satisfactorily with all of the minor revisions that I raised in my previous review.

Proposed major revision

In relation to the one major revision that I proposed, which was concerned with the impact of variations in sample size between farms, the authors presented a clear argument in their response for why they do not regard this as being a problematic issue. The technical details of their argument seem sound, but I'm a bit concerned about the plausibility of the assumptions underpinning it - if I have understood correctly, the argument relies on the assumption that annual yields for individual farms are i.i.d. (independent and identically distributed)? In practice, though, wouldn't we expect correlation between years within a farm, and between farms within a year, with the result that an analysis of individual annual farm yields would conventionally contain random effects for both "year" and "farm" - i.e. would not assume that annual farm-level yields were i.i.d.?

Does the technical argument for why variations in sample size do not alter the underlying distribution of maxima still hold if the i.i.d. assumption is violated, or is the contention of the authors that the i.i.d. assumption is reasonable in that context of these data? In either case, I think some justification for this is needed.

Answer: The referee is correct in pointing out that our technical argument rests on the assumption of independent and identically distributed yields across time and space. This argument was put together to answer the referee's major comment, which read "In situations where a farm has data over multiple years, the authors have selected the maximum yield for that farm, across all years, to go into the analysis. Does this approach not create potential for bias, since the maximum yield per farm will tend to be positively related to the number of years used in constructing the maximum value?"

In our context where only ten years of data are available, evaluating precisely the amount of spatial and/or temporal dependence in the data is a very difficult task. Modelling spatial dependence would require a careful model of the physical processes underpinning wheat growth; in addition to the level of agricultural input, a sensible comprehensive model would need to consider the local chemical composition of soil, which is very difficult to gather and clearly not a simple function of location, as well as (at least!) the behaviour of rainfall, temperature and solar radiation across the regions of interest. State-of-the-art models, such as the model of Kern *et al.* (2018), are typically regression models with a view on modelling average yield levels as a function of physical covariates. As a first step in the analysis of high yields, which to the best of our knowledge has not been tackled so far in the literature, it is, we believe, advisable to start by a simpler model. This is why we contend that a model where yields are considered spatially independent is a reasonable first step in the exploration of the statistical behaviour of high yields.

We could of course propose a theoretical answer to the question of evaluating the influence of alternative ways to data selection (for instance, randomly selecting a single year of data for each farm) on right endpoint estimation along the lines of the reply we gave in the first round of revision. This would not, however, deal with the referee’s concern about dependence between years. To address this concern, we decided to adopt alternative approaches to data selection and evaluate their consequences on finite-sample results. To be more specific, we considered the following samples of yield:

- Yields for the year 2009/2012/2015 only (respectively average/poor/good year for yield),
- Yields for a single randomly selected year of data for each farm,
- Maximum yields over a randomly selected block of 5 years of data.

The rationale for trying selections other than randomly picking a single year of data was that we wanted to test whether selecting an average/poor/good year for yield or maximum yields over a randomly selected smaller number of years had significant consequences on our estimates. [That 2009, 2012 and 2015 can be considered “average”, “poor” and “good” years respectively for yield can be inferred from the boxplots given in Figure 1 of the manuscript.] Let us repeat here that our final objective is **to estimate the best possible yield under current growing conditions**. As a consequence, **we are not interested in the evolution of maximum yield through time** and it is thus meaningful to compare these right endpoint estimates: even though the underlying parameters of the generalised Pareto model may well change when data is constructed by taking maxima over several years (if there is temporal dependence), the maximum possible value of yield, which is our target here, stays unaffected. Consequently, as far as right endpoint estimation is concerned, there is no potential in creating bias by aggregating data across years; in fact, a sample of such aggregated data can intuitively be expected to induce better-performing estimators of the right endpoint, since the values in the data thus created will mechanically tend to be higher than those of single-year data and so closer to our target.

Numerical results are reported in Table 1, and illustrations of the finite-sample behaviour of the associated extreme value index estimator are given in Figures 1 and 2 along with the selected sample fraction in each case. It can be seen in this table that the obtained endpoint estimates for years 2009 and 2012 lie outside the confidence interval for the endpoint calculated using our original construction of the data. This, in our view, should be expected because 2009 and 2012 were respectively average and poor years for yield, and a selection of yield data that is not consistent with our objective of inferring the best possible yield cannot be expected to give sensible results. This is just as in the problem of estimating records in athletics: to do so, it makes sense to consider data made of the best performances of the best athletes (see Einmahl and Magnus, 2008), rather than work on the best performances of average or poor athletes. The estimate based on a single, randomly chosen year of data is also markedly lower than our initial estimate (although perhaps not significantly so at the 95% level). By contrast, the obtained endpoint estimates for year 2015 or for the data made of the best yields across 5 randomly selected years are in line with our initial estimate (if slightly higher but not significantly so), thus illustrating that if an effort is made to select data relevant to our purposes, the actual selection method does not have a strong influence on maximum yield estimates. We therefore argue that the way we selected our data is reasonable.

To further convince the reader that this is the case, and that indeed alternative sensible ways of constructing the data could have been adopted without significantly affecting the results, we have included in a Supplementary Material document the theoretical argument we outlined in the first round of revision, with an appropriate warning about its limitations regarding spatial and temporal dependence, as well as these new analyses based on alternative data selection.

Year	n	k	t	Shape estimate $\hat{\gamma}$	Scale estimate $\hat{\sigma}$	$\hat{x}^* = t - \hat{\sigma}/\hat{\gamma}$
All years combined	1536	250	10.69	-0.11 (-0.22, 0.00)	0.76 (0.65, 0.91)	17.60 (14.02, 23.75)
2009 only	676	140	9.44	-0.22 (-0.35, -0.096)	0.85 (0.70, 1.08)	13.24 (12.31, 14.86)
2012 only	721	250	7.16	-0.17 (-0.28, -0.07)	1.09 (0.93, 1.29)	13.41 (11.88, 16.40)
2015 only	674	100	10.99	-0.095 (-0.27, 0.08)	0.73 (0.57, 0.99)	18.62 (14.02, 31.40)
One randomly chosen year per farm	1536	200	9.96	-0.22 (-0.33, -0.12)	0.98 (0.83, 1.19)	14.32 (13.28, 15.88)
Five randomly chosen years combined	1314	230	10.69	-0.093 (-0.21, 0.02)	0.74 (0.63, 0.89)	18.59 (14.02, 27.56)

Table 1: Maximum yield level estimates \hat{x}^* for our original data set and the data selected in alternative ways, along with a summary of sample sizes, threshold choices, shape estimates $\hat{\gamma}$ and scale estimates $\hat{\sigma}$. Numbers in brackets next to shape, scale and maximum yield estimates represent approximate 95% confidence intervals.

Figure 1: ML estimates of γ , for years 2009 (top left), 2012 (top right), and 2015 (bottom) only.

Figure 2: ML estimates of γ , for data constructed by choosing a single randomly selected year for each farm (top) and the maximum yield over a randomly selected block of 5 years of data (bottom).

Reply to the comments of Referee 3

Thank you very much for your suggestions and comments. The revised version of the manuscript takes all of them into account. For your convenience we incorporate your comments in italics followed by our reply. Modified parts in the revised version of the manuscript are marked in red.

Overview and general comments

It is clear that the authors have attempted to carefully revise and respond to the questions posed by all referees. However, I cannot say I am happy with the revision and the methods the authors tried to implement. All referees have tried to help so that the paper implements well founded statistical principles that effectively overcome almost all problems raised in the analysis of the data. The revised version appears to take a strong stance against profile likelihood intervals and suggests a series of modifications to confidence intervals based on asymptotic normality so that problems do not appear anymore (e.g., lower end point of interval being less than observed maximum in the sample, uncertainty in the upper tail not accounted!, etc.). What is even more unsettling is that the paper suggests evidence for bounded upper tail in crop yield but this seems to be “shown” by, effectively, imposing such an assumption, i.e., by using confidence intervals that are bound to be bounded to the right. Given the author’s strong prior beliefs on the tail of crop yield, it would seem more natural to follow a Bayesian approach and carefully use prior knowledge rather than implementing ad hoc procedures. I include my comments below.

Major

The authors are correct in pointing out that profile likelihood based confidence intervals reflect the uncertainty on the type of the upper tail so that when the return period increases, the profile likelihood becomes flatter. One further strength that hasn’t been pointed out is that profile likelihood based intervals do not result in erroneous lower endpoints, an issue authors decided to address with an ad hoc procedure that involves an adjustment of the endpoint. What is even more unsettling is the argument against using profile likelihood based intervals: the authors suggest in their response that “it is very unlikely that an infinitely large yield is physically and biologically possible... Clearly the method fails here to produce a confidence interval that is bounded to the right, and it is arguable whether it gives a satisfactory numerical result at all”. This argument falls short in several respect as, for example, one can argue in exactly the same way for any physical process. Take rainfall as an example. No matter how much water is evaporated and precipitated, the amount of rainfall will always be finite. However, I have never seen any analysis suggesting intervals bounded to the right be used.

Answer: It is indeed the case that, when calculated for the right endpoint, profile likelihood based intervals have lower bounds larger than the maximum value in the sample. In this sense, they are not erroneous, although one has to remember that profile likelihood intervals are based on (i) the generalised Pareto approximation being accurate for high yield levels, for the constructed likelihood to be meaningful and (ii) a chi-square approximation for the so-called deviance statistic. Strictly speaking, there is therefore no guarantee that profile likelihood based intervals are more correct than asymptotic confidence intervals based on a Gaussian approximation. Furthermore, it is arguable that the practical application we are considering here also requires a sensible calculation of the upper bound of the confidence interval: such an upper bound gives a sense of how much improvement can be brought to currently observed

yield levels, bearing in mind that unbounded yields are physically and biologically impossible. As we explain further in our reply to the referee’s second comment, we agree that “infinitely” was improper terminology and should be replaced by a word such as “arbitrarily”, so as to convey that there is no finite upper bound on the values a heavy-tailed distribution generates. In the rainfall example suggested by the referee, a heavy-tailed model thus makes perfect sense, as there is no obvious physical upper bound on the amount of rain that can fall in a given time period over a given region. Wheat yield per hectare, on the contrary, is strongly constrained by the weight and volume of soil in which the crop grows, therefore making a model with negative shape parameter sensible. We would finally disagree on the comment that “one can argue in exactly the same way for any physical process”: contrary to rainfall, temperature tends to be modelled by distributions with a finite upper bound.

We did and still do recognise that profile likelihood based confidence intervals are an important part of the applied extreme value toolbox. In particular, contrary to what the referee writes in the above comment, the revised version of the manuscript did not take a strong stance against profile likelihood intervals: it actually did not criticise them at all, and instead suggested a modification of the asymptotic confidence interval based on the Gaussian approximation. Our response letter to referees, meanwhile, did include a criticism of profile likelihood intervals on this specific example, but certainly did not generalise that criticism to other types of applications. With this in mind, it is clear that there is a disagreement between us and the referee on the use of profile likelihood on the sample of data analysed in the present paper. We have therefore decided to include the profile likelihood calculations and graphs related to the full sample of yield in a Supplementary Material document; they are referred to on page 7 of the revised version of the manuscript and reproduced in Figures 3 and 4 below. This will, we believe, help the reader make an evidence-based decision of which of the two calculations (profile likelihood or Gaussian approximation) is preferable, depending on their objectives and statistical preferences.

Observing an infinitely large yield would only be possible had the underlying distribution had an atom at infinity which no extreme value model entertains. So, after reading the author’s comment, it is clear that there is a confusion about what $\xi > 0$ means.

Answer: Our point was that an extreme value index $\xi > 0$ for the distribution of yield implies that no upper bound can be given on yield: see lines 154-155, where we wrote “the theoretical possibility of a heavy tail which would imply an unbounded maximum yield”. This, we argued on lines 151-165, is biologically and physically impossible. We agree with the referee that “infinitely” on line 164 was not the correct word to use, and we replaced it by “arbitrarily” in the revised version.

In my previous report I recommended return level plots are presented instead of point and interval estimates of extreme value model parameters. Of course, the maximum of the distribution is a return level but a return level plot shows the full tail by plotting return levels against different return periods.

Answer: The referee’s related comment in his/her first report was that “Reporting estimates of location, scale or tail indices obtained from an extreme value analysis is cumbersome. Instead, it is better practice to include return levels and their associated uncertainty as these have a direct interpretation and facilitate readability by non-experts.” Return levels do indeed facilitate readability. We would argue though that an important question when reporting return levels is whether they are relevant both within the field of study and for communication outside this field. In hydrology, for instance, the estimation of once-in- N years return levels is of interest to governments: the Dutch government is interested in the magnitude of a once-in-10,000 years flood so as to be able to build and maintain dikes protecting areas below sea level against such

floods (see the Preface of de Haan and Ferreira, 2006). In this kind of situation, reporting return levels makes sense, because governments need to balance cost and safety while having access to results that government workers, who may have some training for handling data but will overwhelmingly be non-statisticians, can easily read. In the application we are interested in, we argue that the focus of farmers and government consists of mean/median yield (allowing a rough prediction of whether wheat farming is viable and of total output across a country within a year) and maximum possible yield (to establish how far current production levels are from optimal levels). We do not think that a once-in-10 years yield level, say, is relevant in practical applications, as no farm or government is going to take any decision regarding production or financial support measures based on an agricultural output reached once on average in a decade. This is why we choose not to report return levels in the manuscript. As we mentioned in our first reply to the referee, it is indeed the case that we briefly report parameter estimates early in the Results section and in Table 1; this should be seen as a way to help the reader unfamiliar with extreme value analysis understand that our maximum yield estimates are reached by combining the threshold, scale and shape parameter estimates.

The profile likelihood based intervals in the report to referees suggest there are cases where there isn't sufficient evidence to claim a bounded upper tail as 0 is contained in the interval. Since stratification reduces sample size and increases the uncertainty in the estimates, I would suggest authors think in more detail about how to better model the physical processes so that information is pooled (e.g., across space or otherwise) a point that was raised by another referee. The authors decided to suggest that this goes beyond the scope of the present paper which in its current form, simply fits extreme value models to strata and chops interval estimates to claim evidence for bounded tails.

Answer: It is indeed correct that Referee 1 raised the possibility of creating a different model to produce lower standard errors. We did try versions of such models where scale or shape is kept constant across regions: as we showed in our reply to this referee on the example of a model with a constant scale parameter, this results in poorer model fits and makes us prefer the stratified model we are currently using.

It is also correct, as we acknowledged in our reply to Referee 1, that a model where the parameters depend smoothly on the covariates, or a model better accounting for how the underlying physical processes behave (as mentioned here by the referee), is worth pursuing. The situation we are working in, however, is more complex than the modelling of standard physical processes such as rainfall and temperature: in addition to the level of agricultural input, a sensible comprehensive model would need to consider the local chemical composition of soil, which is very difficult to gather and clearly not a simple function of location, as well as (at least!) the behaviour of rainfall, temperature and solar radiation across the regions of interest. State-of-the-art models, such as the model of Kern *et al.* (2018), are typically regression models with a view on modelling average yield levels as a function of physical covariates. These models are not written at the fine level of detail we would require for our purposes; in particular, even in this arguably much simpler case of modelling average rather than high yields, they do not seek to provide a parametric model for yield whose parameters would depend on biological and/or physical covariates smoothly and in a specified way. A semiparametric model, in the spirit of those of Chavez-Demoulin and Davison (2005) and Chavez-Demoulin *et al.* (2016), is in theory possible to solve the problem of the unknown form of dependence of model parameters on covariates, but they will necessarily require the estimation of a nonparametric component on which the uncertainty is likely to offset any gain in pooling together the data across regions. This is on top of the obvious difficulties in estimating such models, be it by the choice of the number of knots in spline models or of the bandwidth in smooth backfitting-type procedures. More generally, as we mention in the Introduction of the manuscript, the focus

of the applied statistical literature on understanding agricultural yield variability has typically been to estimate average yield levels and understand the relationships that drive them using central, rather than extreme, statistical methodology. As a first step in the analysis of high yields, which to the best of our knowledge has not been tackled so far in the literature, it is, we believe, advisable to start by a simpler model highlighting what kind of distribution would be appropriate to use and giving an idea of differences across regions and input use, just as we do in the present manuscript. The refinement of our analysis into a more precise model is definitely worth pursuing in future work on this topic, as we mention in the final lines of our “Discussion and conclusions” section.

Minor

globally replace “Equation” by “equation”.

Answer: Done.

pg 6. Replace “Gaussian confidence interval” by “approximate confidence interval based on asymptotic normality”.

Answer: Done.

pg 6. “This allows approximate confidence intervals for” replace by “This allows approximate confidence intervals based on asymptotic normality for”.

Answer: Done.

pg 6. “for our maximum yield estimates”. Replace by “for the true maximum yield”. The interval is an estimator of the “true quantity” of interest, not of the point estimate itself.

Answer: Done.

Figure 3: Full yield data set of size $n = 1536$, graph of the profile likelihood function for the shape parameter γ . The two intersections of the concave full line with the second-from-top horizontal full line define the 95% profile likelihood confidence interval for γ . The vertical dashed line is the line $\gamma = 0$.

Figure 4: Full yield data set of size $n = 1536$, graphs of the profile likelihood functions for return levels. Top left: return level with exceedance probability $10/n$, top right: return level with exceedance probability $1/n$, bottom left: return level with exceedance probability $1/(10n)$, bottom right: return level with exceedance probability $1/(100n)$. In each graph, the two intersections of the concave full line with the second-from-top horizontal full line define the 95% profile likelihood confidence interval.

References

- Chavez-Demoulin, V. and Davison, A.C. (2005). Generalized additive modelling of sample extremes, *Journal of the Royal Statistical Society: Series C* **54**(1): 207–222.
- Chavez-Demoulin, V., Embrechts, P. and Hofert, M. (2016). An extreme value approach for modeling operational risk losses depending on covariates, *Journal of Risk and Insurance* **83**(3): 735–776.
- de Haan, L. and Ferreira, A. (2006). *Extreme Value Theory: An Introduction*, Springer.
- Einmahl, J.H.J. and Magnus, J.R. (2008). Records in athletics through extreme-value theory, *Journal of the American Statistical Association* **103**(484): 1382–1391.
- Kern, A., Barcza, Z., Marjanović, H., Árendás, T., Fodor, N., Bónis, P., Bognár, P. and Lichtenberger, J. (2018). Statistical modelling of crop yield in Central Europe using climate data and remote sensing vegetation indices, *Agricultural and Forest Meteorology* **260–261**: 300–320.